



# From fibrous plant residues to mineral-associated organic carbon – the fate of organic matter in Arctic permafrost soils

Isabel Prater[1], Sebastian Zubrzycki[2], Franz Buegger[3] Lena C. Zoor-Füllgraff[1], Gerrit Angst[4], Michael Dannenmann[5], Carsten W. Mueller[1]

[1]Technical University of Munich, Research Department Ecology and Ecosystem Management, Soil Science, 85354 Freising, Germany

[2]University of Hamburg, Center of Earth System Research and Sustainability, School of Integrated Climate System Sciences, 20146 Hamburg, Germany

[3]Helmholtz Zentrum München, Institute of Biochemical Plant Pathology, 85764 Neuherberg, Germany

[4]Biology Centre of the Czech Academy of Sciences, Institute of Soil Biology & SoWa Research Infrastructure, 370 05 České Budějovice, Czech Republic

[5]Karlsruhe Institute of Technology, Institute for Meteorology and Climate Research, Atmospheric Environmental Research (IMK-IFU), 82467 Garmisch-Partenkirchen, Germany

*Correspondence to*: Isabel Prater (i.prater@tum.de)

**Abstract.** Permafrost-affected soils of the Arctic account for 70 % or 727 Pg of the soil organic carbon (C) stored in the permafrost region and therefore play a major role in the global C cycle. Most studies on the budgeting of C storage and the quality of soil organic matter (SOM) in the northern circumpolar region focus on bulk soils. Thus, although there is a plethora of assumptions regarding differences in terms of C turnover or stability, only little knowledge is available on the mechanisms stabilizing organic C in Arctic soils besides impaired decomposition due to low temperatures. To gain such knowledge, we

investigated soils from Samoylov Island in the Lena River Delta with respect to the composition and distribution of organic C among differently stabilized SOM fractions. The soils were fractionated according to density and particle size to obtain differently stabilized SOM fractions differing in chemical composition and thus bioavailability. To better understand the chemical alterations from plant-derived organic particles in these soils rich in fibrous plant residues to mineral-associated SOM, we analysed the elemental, isotopic and chemical composition of particulate OM (POM) and clay-sized mineral-

associated OM (MAOM). We demonstrate that the SOM fractions that contribute with about 17 kg C m$^{-3}$ for more than 60 % of the C stock are highly bioaccessible and that most of this labile C can be assumed to be prone to mineralization under warming conditions. Thus, the amount of relatively stable, small occluded POM and clay-sized MAOM that account currently with about 10 kg C m$^{-3}$ for about 40 % of the C stock will most probably be crucial for the quantity of C protected from mineralization in these Arctic soils in a warmer future. Using $\delta^{15}$N as proxy for nitrogen (N) balances indicated an important

role of N inputs by biological N fixation, while gaseous N losses appeared less important. However, this could change, as with about 0.4 kg N m$^{-3}$ one third of the N is present in bioaccessible SOM fractions, which could lead to increases in mineral N cycling and associated N losses under global warming. Our results highlight the vulnerability of SOM in Arctic permafrost-affected soils under rising temperatures, potentially leading to unparalleled greenhouse gas emissions from these soils.

## 1 Introduction

For several millennia, organic matter (OM) accrued in the remote soils of the Arctic and only recently researchers started to increasingly understand the importance of these cold soils for the global carbon (C) cycle and, thus, global climate (Ping et al., 2015). Estimates on the northern circumpolar soil organic carbon (SOC) stock within the first meter vary between 445 and 496 Pg (Tarnocai et al., 2009, Hugelius et al., 2014). These C-rich soils are changing from a C sink to a source due to global warming (Oechel et al., 1993; Parmentier et al., 2017). In a warming Arctic, C is lost both via carbon dioxide and methane

emissions and by lateral transport with water (Plaza et al., 2019). The C that is released from permafrost-affected soils due to



anthropogenically accelerated thawing is assumed to further enhance global warming and thus trigger additional C release from permafrost, a phenomenon known as permafrost C feedback (Davidson and Janssens, 2006; Schuur et al., 2015).

An analysis of soils from ten North American ecosystem types reaching from tropical forests to Arctic tundra demonstrated a pronounced longer turnover time for soil organic matter (SOM) in cold regions in comparison to other climate regions as the

C stabilization mechanisms clearly differ (Frank, Pontes and McFarlane, 2012). In temperate soils, the main drivers for SOC sequestration are spatial inaccessibility (occlusion in soil aggregates), binding to mineral particles (organo-mineral associated OM), and intrinsic chemical recalcitrance of the OM itself (Six et al., 2002; von Lützow et al., 2006). Besides these specific mechanisms, environmental factors like waterlogging and low temperatures inhibit the turnover of OM in cold regions (Oades, 1988), with cryoturbation additionally supporting the conservation of SOM at greater soil depth and thus in the permafrost

(Kaiser et al., 2007). These abiotic mechanisms fail as soon as permafrost collapses, which leads to an increased decomposition of OM (Turetsky, 2004; Plaza et al., 2019).

Already in 1982, Post et al. recognized a considerable variability in C stocks in tundra soils, which illustrates that a more detailed knowledge on the biogeochemical cycling of C in permafrost soil needs to involve more analytical approaches that enable to assess possible mechanisms of C stabilization. Thus, besides the quantification of organic C (OC), there is a growing

number of studies aiming to elucidate the chemical composition of SOM and the processes and mechanisms involved in C cycling and stabilization in permafrost-affected soils (i.a. Torn et al., 2013; Mueller et al., 2015; Strauss et al., 2017; Jongejans et al., 2018; Kuhry et al., 2019).

The Arctic is strongly affected by climate change with an increase in surface temperatures during the last two decades that is more than twice the global average (Meredith et al., 2019). With ongoing warming, the active layers in cold regions deepen

and thus, microbial activity changes and the accessibility and bioavailability of OM in hitherto frozen soil layers increases (Mackelprang et al., 2011; Hultman et al., 2015). Depolymerization and ammonification as well as nitrification of the long sequestered organic nitrogen (N) might also enhance mineral N availability in these permafrost-affected soils, leading to increased emissions of the highly potent greenhouse gas nitrous oxide (Elberling et al., 2010; Wilkerson et al., 2019). The importance of mechanisms restricting SOM decomposition in permafrost soils will possibly shift from climatic stabilization

(Schmidt et al., 2011) to spatial inaccessibility and association with minerals (Harden et al., 2012; Mueller et al., 2015) with widely unknown consequences for the C stored in these soils. Several studies estimated the vulnerability of C in permafrost soils to microbial decay from the chemical composition of bulk SOM (i.a. Herndon et al., 2015; Strauss et al., 2017; Tesi et al., 2016; Weiss and Kaal, 2018; Wild et al., 2016; Xue et al., 2016; Zimov et al., 2006). Yet, as SOM represents a continuum of a range of materials of different composition, from fresh plant litter to highly altered compounds (Lehmann and Kleber,

2015) ruled by different stabilization regimes, the investigation of bulk SOM alone is insufficient. The use of more sophisticated approaches, separating SOM into different fractions, allows for a more detailed understanding of the stabilization mechanisms in soil (Golchin et al., 1994). So far, only few studies (i.a. Dao et al., 2018; Diochon et al., 2013; Dutta et al., 2006; Gentsch et al., 2015; Höfle et al., 2013; Mueller et al., 2015; Xu et al., 2009) used fractionation approaches to investigate the distribution and composition of OM pools in permafrost-affected soils (Ping et al., 2015), most of them focusing on the

composition of specific fractions or using incubation experiments. We used a physical fractionation approach to separate light organic particles and OM associated with minerals, i.e. particulate OM (POM; dominated by bits and pieces of plant and to a lesser extent microbial residues) and mineral-associated OM (MAOM) to gain detailed insights into the chemical composition and stabilization mechanisms of SOM under present conditions of Cryosols in the Siberian Lena delta.





## 2. Methods

### 2.1 Site characteristics and soil sampling

Samoylov Island (72° 22' N, 126° 30' E) is located in one of the main channels of the Lena River Delta, the largest delta of the Arctic. The island developed during the Holocene and belongs to one of three river terraces. While the western third of the island consists of an active floodplain, the eastern part is covered by ice-wedge polygonal tundra that is typical for this terrace (Boike et al., 2013). Located at 10 to 16 m a.s.l., the Holocene river terrace is rarely flooded and its plant cover represents the characteristic wet sedge tundra vegetation (Zubrzycki et al., 2013). This terrace has recently been reported to be covered by about 40 % non-degraded polygonal tundra, 40 % collapsed polygons, slopes, and water bodies, and 20 % of polygons that show different stages of degradation (Kartoziia, 2019). On the island, active layer thickness varies around 50 cm and the thawing period lasts approximately 129 days (Boike et al., 2013). The climate is arctic and the 30-year mean (1961-1990) of the closest meteorological station in Tiksi, about 110 km southeast, shows a mean annual air temperature of -13.5° C with a large amplitude between warmest (around 8° C in July and August) and coldest (around -32° C in January) months (Roshydromet, 2019). Precipitation is low on Samoylov Island and, due to the different geographic setting within the river delta, with a mean of 125 mm a$^{-1}$ markedly lower than the 323 mm a$^{-1}$ measured in Tiksi (Boike et al., 2013; Roshydromet, 2019).

We drilled four intact soil cores from ice-wedge polygon centers (Fig. 1; Boike et al., 2012) in April 2011 and May 2013 using a Snow-Ice-Permafrost-Research-Establishment coring auger (Jon's Machine Shop, Fairbanks/USA) with a length of 1 m and a diameter of 76 mm with a STIHL BT 121 engine power head (Andreas Stihl AG & Co. KG, Waiblingen/Germany). A detailed description of the study area and the sampling of the soil cores can be found in Zubrzycki et al. (2013).

All bulk soil samples were slightly acid with lowest pH values of 4.9 and highest of 6.6, electric conductivity ranged from 66 to 240 µS cm$^{-1}$ with a mean of 115 µS cm$^{-1}$ and bulk density varied from 0.2 to 0.9 g cm$^{-3}$ around a mean of 0.5 g cm$^{-3}$.

### 2.2 Geochemical properties of bulk soils, physical soil fractionation and chemical analyses of fractions

We separated the drilled cores according to visible mineral soil horizons in frozen condition and subsequently thawed and dried them at 40° C in an oven. Our analyses focused on selected layers only, as shown in Table 1.

The bulk soils were fractionated according to density and particle-size, following the approach described by Mueller and Koegel-Knabner (2009). Due to the high amount of fibrous material in these Cryosols, some modifications of the procedure were necessary to yield mechanistically different SOM fractions. We unclenched 15 to 20 g – depending on the available amount of sample material – of each soil sample by forceps and gently saturated them with a sodium polytungstate solution with a density of 1.8 g cm$^{-3}$ by slowly adding the salt solution with a pipette. After 12 hours to ensure a complete and gentle saturation, the floating free POM (fPOM, not embedded in stable aggregates, cf. Golchin et al., 1994) was collected using a vacuum system. The removal of the floating fPOM was repeated twice to ensure a high recovery and the obtained fraction was subsequently washed over a sieve of 20 µm mesh size to remove excessive salt. Due to the highly fibrous nature of the fPOM, the washing step also yielded fine mineral particles, which adhered to the fPOM fibers. As the C and N contents and C/N ratios of this mineral material were in the exact same range of the clay-sized MAOM fraction, we added it mathematically to this fraction for the calculation of the C stock. To separate occluded POM fractions (oPOM, incorporated in water-stable aggregates, cf. Golchin et al., 1994) from MAOM, the residual samples were subjected to ultrasonication (Bandelin Sonoplus HD2200, Berlin/Germany) using a calibrated (Graf-Rosenfellner et al., 2018) energy input of 300 J ml$^{-1}$ after the fPOM removal. On the lines of the fPOM fractions, oPOM was withdrawn using a vacuum system and washed salt-free over a sieve of 20 µm mesh size by repeated washing until the EC dropped below 2 µS. During the washing of the oPOM, we obtained the small oPOM (oPOMs) fraction representing a fine particulate light OM (Mueller et al., 2015, 2017). The remaining heavy residues, constituting the MAOM, were separated by wet sieving and sedimentation to obtain coarse/medium sand (>200 µm),



fine sand (63-200 µm), coarse silt (20-63 µm), medium silt (6.3-20 µm) and fine silt/clay (<6.3 µm, further referred to as the clay-sized MAOM fraction). All SOM fractions were analyzed for total C and N contents in duplicate by dry combustion (EuroVector EuroEA3000 Elemental Analyser, Pavia/Italy). After the analyses of each sample, for better clarity for the reader, C and N contents were calculated for the combined sand- and silt-sized fraction per each bulk soil sample. Due to the absence of carbonates (see pH values in Table 1), total C represents OC. Coarse fractions >20 µm were ball milled and homogenized

prior to C and N measurements. The bulk soil C and N contents were calculated from the sum of the physical fractions; C and N stocks for the SOM fractions were also calculated and overall C and N stocks projected to 1 m soil depth. The mass recovery rate after fractionation was >90 % in all samples. In addition, to reveal the microscale structure and illustrate possible source materials (microbial vs. plant origin) scanning electron microscope (SEM) images (JSM-7200F, JEOL, Freising/Germany) were obtained for representative POM fractions.

### 2.3 Stable isotope measurements

The abundance of $^{15}$N and $^{13}$C of POM and clay-sized MAOM fractions were determined using an isotope ratio mass spectrometer (Delta V Advantage, Thermo Fisher, Dreieich/Germany) coupled to an elemental analyzer (EuroEA, Eurovector, Pavia/Italy). A lab standard (acetanilide) was used as a standard for every sequence in intervals and different weights as well to quantify isotope linearity of the system. The standard itself was calibrated against several suitable international isotope

standards from the International Atomic Energy Agency (IAEA, Vienna/Austria) for both isotopes. Final correction of isotope values was achieved with several international isotope standards and other suitable laboratory standards that cover the range of $\delta^{15}$N and $\delta^{13}$C results. Results are given in delta values relative to air-N$_2$ for $^{15}$N and relative to Vienna Pee Dee Belemnite (V-PDB) for $^{13}$C (Werner and Brand, 2001)

### 2.4 $^{13}$C Nuclear Magnetic Resonance Spectroscopy

We subjected all fPOM, oPOM, oPOMs and selected clay-sized MAOM fractions to $^{13}$C cross-polarization magic angle spinning (CP-MAS) NMR spectroscopy (Bruker DSX 200 spectrometer, Billerica/USA). The $^{13}$C NMR spectra were recorded at 6,800 Hz with an acquisition time of 0.01024 s. During a contact time of 1 ms, a ramped $^1$H pulse was applied to avoid Hartmann-Hahn mismatches. We executed measurements in 7 mm zirconium dioxide rotors with a delay time of 1.0 s for large POM fractions (fPOM and oPOM) and a reduced delay time of 0.4 s for oPOMs and clay-sized MAOM fractions. The acquired

number of scans (NS) varied according to the examined fractions and the available sample material. For most of the large POM fractions, a NS between 3,000 and 10,000 provided sufficient signal-to-noise ratios, while most of the oPOMs and clay-sized MAOM fractions required a NS of at least 10,000. Tetramethylsilane was equalized with 0 ppm as reference for the chemical shifts. The spectra were integrated in different chemical shift regions according to Beudert et al. (1989) with slight adjustments according to Mueller and Koegel-Knabner (2009): -10 to 45 ppm (alkyl C), 45 to 110 ppm (O/N alkyl C), 110 to

160 ppm (aromatic C) and 160 to 220 ppm (carboxyl C), spinning sidebands were included. Based on these integrated shift regions, we calculated the ratio of alkyl C and O/N alkyl C (a/o-a ratio) as a proxy for the degree of decomposition of plant residues according to Baldock et al. (1997). Furthermore, we calculated the ratio of the integrated chemical shift regions 70 to 75 ppm (O alkyl C of carbohydrates) and 52 to 57 ppm (methoxyl C of lignin) according to Bonanomi et al. (2013), which provides another proxy for the decomposition stage of plant residues in relation to fresh plant source material (further referred

to as 70-75/52-57 ratio). To translate the NMR spectra into OM compound classes (carbohydrate, protein, lignin, lipid, carbonyl) we fitted the NMR data using the molecular mixing model (MMM) developed by Nelson and Baldock (Baldock et al., 2004; Nelson and Baldock, 2005). For the MMM fitting, we utilized the following chemical shift regions: 0 to 45 ppm, 45 to 60 ppm, 60 to 95 ppm, 95 to 110 ppm, 110 to 145 ppm, 145 to 165 ppm and 165 to 215 ppm. We applied the five component MMM (without char) with N:C constraint.





**2.5 Statistics**

We plotted C/N ratios and C and N concentrations against the N and C stable isotope ratios of SOM fractions with Microsoft Excel 2016 to detect correlations. The R software, RStudio and Rcmdr (with the FactoMineR plugin) were used for Principal Component Analysis (PCA), correlation matrices and the compilation of plots (Lê et al., 2008; RStudio Team, 2016; R Development Core Team, 2017). We used PCA and correlation matrices to find correlations in different SOM fractions (fPOM,

oPOM, oPOMs, clay-sized MAOM). For this purpose, we analyzed C and N contents, decomposition proxies (C/N ratio of bulk soils and of SOM fractions, a/o-a ratio, 70-75/52-57 ratio), stable isotopes, and the results from the MMM.

**3. Results**

**3.1 Biogeochemical bulk soil properties and distribution of SOM fractions**

The bulk soil C contents over all cores and depth layers varied between 31.6 and 144.0 mg g$^{-1}$. The content of total N ranged
from 1.3 to 6.8 mg g$^{-1}$ for all cores and depth layers. While the C/N ratios ranged between 23 and 38 in three of the four cores, the values of the bulk soils of the fourth core were markedly lower (Table S1). The soil C stocks (projected to 1 m soil depth) ranged between 20.4 and 31.4 kg C m$^{-3}$ with a mean of 27.5±11.8 kg C m$^{-3}$, the N stocks varied between 0.7 and 1.9 kg N m$^{-3}$ with a mean of 1.2±0.6 kg N m$^{-3}$ (Table 1).

The mass distribution of POM fractions varied throughout all depth layers with proportions between 10.6 and 295.0 mg g$^{-1}$
(fPOM), between 3.0 and 71.7 mg g$^{-1}$ (oPOM) and between 3.9 and 267.2 mg g$^{-1}$ (oPOMs). Especially core 3 and 4 showed larger amounts of fPOM and oPOM material at greater depth in between layers dominated by MAOM (Table S1). The MAOM fractions ranged between 37.2 and 244.5 mg g$^{-1}$ (clay-sized), between 182.4 and 479.3 mg g$^{-1}$ (silt-sized) and between 79.0 and 591.5 mg g$^{-1}$ (sand-sized).

**3.2 Elemental composition of SOM fractions**

The highest C contents were detected in the fPOM and oPOM fractions, with values ranging from 196.3 to 425.5 mg g$^{-1}$ C for the fPOM and from 368.4 to 449.1 mg g$^{-1}$ C for the oPOM fractions. Due to the highly fibrous structure of these Cryosols rich in plant residues, fractionation was challenging for some of the samples, leading to one outlier within the fPOM fractions and four outliers within the oPOM fractions. We defined outliers as the measurements laying outside the boxplots' whiskers, thus values lower than 1.5 times the interquartile range below the lower quartile and values higher than 1.5 times the interquartile
range above the higher quartile (Fig. 2). We excluded these fractions from further calculations as we assume that they point to mineral particles, which we were not able to separate fully from the very fibrous POM structures. The C content of the oPOMs fractions ranged between 61.4 and 344.8 mg g$^{-1}$ C, the C contents of the clay-sized MAOM fractions between 51.5 and 117.9 mg g$^{-1}$ C, while silt- and sand-sized MAOM fractions showed the lowest C contents (Fig. 2).

Results for the N content were 5.0 to 19.5 mg g$^{-1}$ N for fPOM fractions, 3.4 to 23.7 mg g$^{-1}$ N for oPOM fractions and slightly
higher for oPOMs fractions with 4.6 to 26.4 mg g$^{-1}$ N. The N contents of the clay-sized MAOM fractions ranged between 3.8 and 10.1 mg g$^{-1}$ N, while silt- and sand-sized MAOM fractions contained markedly less N (Fig. 2). Large POM fractions (fPOM and oPOM) showed a wide variation of C/N ratios with values between 22 and 76 for fPOM and between 18 and 113 for oPOM. The values of the oPOMs fractions were clearly lower and had less variability with 13 to 25, while clay-sized MAOM fractions ranged between 11 and 16. Lowest C/N ratios were present in silt- and sand-sized MAOM fractions with 8
to 12 and 6 to 19, respectively. Large POM fractions had not only the widest C/N ratios compared to the oPOMs and mineral-associated OM within each soil layer, but also showed the largest variation (Table S2, Fig. 3).

The contribution of C and N weighted for the amount of each specific SOM fraction per soil layer showed a great variance in the amount of C and N stored either as POM or MAOM. For C, this ranged between 211.5 and 807.0 mg C per g bulk soil for





the large POM fractions (fPOM and oPOM), between 13.7 and 479.7 mg C per g bulk soil for the oPOMs, whereas the clay-
sized MAOM ranged between 59.4 and 431.4 mg C per g bulk soil (Table S2).

Over all analyzed soil layers, POM fractions accounted for 80 % of the C stock (22.0±9.2 kg C m$^{-3}$), while the MAOM fractions
accounted for about 20 % (5.5±2.7 kg C m$^{-3}$). Overall, the fPOM fractions dominated the C stock, with 14.0±4.6 kg C m$^{-3}$
representing about half of the total C stock of all analyzed cores and layers. The occluded POM fractions contributed less with
2.6±1.1 kg C m$^{-3}$ (oPOM) and 5.4±3.5 kg C m$^{-3}$ (oPOMs). The share of the clay-sized MAOM fractions in the C stock was
4.6±2.2 kg C m$^{-3}$, while silt- and sand-sized MAOM fractions played only a subordinate (Table 1).

For the N stock, the contribution of the POM fractions sums up to about 60 % (0.7±0.4 kg N m$^{-3}$) and that of the MAOM
fractions to about 40 % (0.5±0.2 kg N m$^{-3}$). The fPOM and oPOM fractions contributed differently to the stock with
0.3±0.1 kg N m$^{-3}$ and 0.1±0.1 kg N m$^{-3}$, respectively. The oPOMs and clay-sized MAOM fractions added similarly to the N
stock with 0.3±0.2 kg N m$^{-3}$ and 0.4±0.2 kg N m$^{-3}$, but also showed the largest variation. Similar to C stocks, silt- and sand-
sized MAOM fractions had a negligible share in the N stocks (Table 1).

Although overall the soil C and N storage was dominated by POM, the distribution of POM- vs. MAOM-related C and N
varied greatly with depth, with some soil layers showing a dominance of MAOM for C and N storage (Table S2).

### 3.3 Isotopic composition of SOM fractions

For POM and clay-sized MAOM fractions, we analyzed the content of stable carbon ($^{13}$C) and nitrogen ($^{15}$N) isotopes. With
respect to $\delta^{15}$N, the values differed little between all examined fractions: fPOM (-0.3 to 1.4 ‰), oPOM (0.2 to 2.4 ‰), oPOMs
(0.0 to 2.9 ‰) and clay-sized MAOM (-0.4 to 3.4 ‰) fractions with the latter showing the highest values (Table S2). With
decreasing C/N ratios, a clear trend towards more negative $\delta^{13}$C and lower $\delta^{15}$N values was demonstrated for all POM fractions
(Fig. 4). As shown by PCA (Fig. 5), $\delta^{15}$N and $\delta^{13}$C showed positive dependencies with the C/N ratios. As the deeper soil layers
of core 4 were clearly dominated by MAOM with a narrow C/N ratio, the overall $\delta^{15}$N (0.7 ‰) was lower compared to the
other three cores.

The $\delta^{13}$C values were similar for all fractions and well within the range of SOM derived from plants with a C$_3$ metabolism
(Sharp, 2007). The range of $\delta^{13}$C values and their variability was similar for fPOM (-31.2 to -25.6 ‰), oPOM (-30.6
to -25.3 ‰), oPOMs (-31.5 to -25.0 ‰) and clay-sized MAOM (-31.8 to -24.1 ‰: Table S2). As for $\delta^{15}$N, also the $\delta^{13}$C values
of the soil material of core 4 differed from those in the other cores showing clearly lower values. Thus, overall the differences
between the cores were larger than the differences between the fractions. Also for the $\delta^{13}$C values, a relation to the C/N ratios
of all fractions was demonstrated. The C/N ratios of the clay-sized MAOM asymptotically approached a limit when plotted
over the $\delta^{15}$N and $\delta^{13}$C, whereas the POM fractions showed a linear increase in the isotope content at higher C/N ratios (Fig. 4).

### 3.4. $^{13}$C NMR – the molecular level

The $^{13}$C CP-MAS NMR spectra of all examined SOM fractions showed dominant peaks in the O/N alkyl C region. The spectra
of both large POM fractions were clearly dominated by the shouldered major peak around 70 ppm and a minor peak around
105 ppm. The integration of the spectra fortified the dominance of O/N alkyl C with about 70 % in the fPOM (n=22) and
oPOM (n=19) fractions (Table 4). In the regions of carboxyl and alkyl C small peaks were present, with only a small hump
being present in the aromatic C region. The differences between the spectra of the fPOM and oPOM fractions (see Fig. S1)
and in their relative composition were only minor, even shoulders and minor side peaks were comparable in the majority of
the samples. In contrast, spectra of the oPOMs (n=23) and clay-sized MAOM (n=10) fractions showed pronounced peaks
around 30 ppm in the alkyl C region and around 170 to 175 ppm in the carboxyl C region. Throughout all samples, there was
a shift from a high percentage of O/N alkyl C in the large POM fractions to a higher percentage of aromatic and alkyl C in
oPOMs and clay-sized MAOM fractions (Table 2).



To get more differentiated information about the degree of decomposition of the OM, we calculated the a/o-a-ratio for the
SOM fractions (Baldock et al., 1997). While fPOM and oPOM fractions revealed identically low values and relatively large
standard deviations with 0.2±0.1, oPOMs and clay-sized MAOM showed clearly higher values with about 0.5. Additional to
the a/o-a-ratio, we applied the 70-75/52-57 ratio (Bonanomi et al., 2013) to the SOM fractions and received results consistent
with the a/o-a-ratios: fPOM and oPOM showed high values, indicating a low degree of decomposition, while oPOMs and clay-
sized MAOM showed very low values. With this ratio, the large POM fractions showed a considerable variance, while the
deviation within oPOMs and clay-sized MAOM was marginal (Fig. 6). Figure 7 illustrates the close relation between the C/N
ratio of the SOM fractions and the NMR-derived decomposition proxies.

By modelling the molecular composition of the SOM fractions using the MMM (Baldock et al., 2004; Nelson and Baldock,
2005), we obtained a clear differentiation between the large POM fractions (fPOM, oPOM) and small oPOM and clay-sized
OM separates. The composition of the fPOM and oPOM fractions was rather similar: the percentage of carbohydrates (about
60 %) was highest and at the same time, the contribution of lipids (about 8 %) was lowest in these fractions (Table 3). Overall,
the composition of both large POM fractions was similar with slightly lower amounts of protein and slightly higher amounts
of carbonyl in oPOM compared to fPOM. The usage of the MMM revealed once more clear differences between the large
POM fractions and oPOMs and clay-sized MAOM. The latter fractions had a lower percentage of carbohydrates (about 40 %),
whereas the percentage of protein and lipids was markedly higher. These fractions differed mainly in the proportion of protein
and lipids, with clay-sized MAOM containing a larger proportion of protein, but a smaller proportion of lipids (Table 3). The
proportion of carbonyl was overall low with high deviations, while the percentage of lignin was rather constant throughout all
four examined fractions.

The PCA executed on the examined fractions showed slight correlation between the abundance of stable isotopes and NMR-
derived decomposition proxies; yet, it confirmed the close relation between fPOM and oPOM and the positioning of oPOMs
between large POM and clay-sized MAOM fractions (Fig. 5). The separation of the large POM fractions and oPOMs fractions
for a second PCA and correlation matrices provided more details on the correlations (Fig. 8). While the first PCA (Fig. 5)
already gave a hint, the correlation matrix demonstrated that in the large POM fractions both $\delta^{15}N$ and $\delta^{13}C$ were slightly
positively correlated with the 70-75/52-57 ratio and negatively correlated with the a/o-a ratio. The positive correlation between
$\delta^{13}C$ and the a/o-a ratio was strong in the oPOMs fractions and the negative correlation between $\delta^{13}C$ and the 70-75/52-57 ratio
was more pronounced, whereas $\delta^{15}N$ was not correlated with the 70-75/52-57 ratio, but negatively correlated with the a/o-a
ratio in the oPOMs fractions.

## 4. Discussion

### 4.1 Permafrost processes determine bulk soil organic matter distribution

The found projected mean C stock of 27.5±11.9 kg C $m^{-3}$ corresponds with those reported in other studies from the Siberian
Arctic (cf. Zubrzycki et al., 2014, where the authors demonstrated values between 6.6 and 48.0 kg C $m^{-3}$ in their overview).
Besides the large amount of sequestered C, a considerable amount of N is stored in permafrost-affected soils. Despite often
named as a decisive factor for plant growth in usually N deficient tundra ecosystems (Weintraub and Schimel, 2005), soil N
stocks strongly dominated by polymeric organic N might not be related to N availability for plants in the form of amino acids
or mineral N. The values for N stocks of permafrost-affected soils reported by other authors (cf. Fuchs et al., 2018; Zubrzycki
et al., 2013, demonstrating N stocks ranging between 1.1 and 2.2 kg N $m^{-3}$) are similar to our results of 1.2±0.6 kg N $m^{-3}$.

The ample range of the bulk soil C/N ratios points to a wide variance in composition and degree of decay of the SOM. The
C/N ratios notably differed both between the single depth layers and the overall cores. The variable bulk soil C/N ratios with
depth can be assigned to the translocation of fresh plant-derived OM by cryoturbation, leading to specific soil layers with
higher C/N ratios (Kaiser et al., 2007; Krueger et al., 2014). Such an incorporation of OM in subsoil is also confirmed by the





high percentage of POM fractions present in these depth increments dominated by fibrous plant residues. Between the analyzed cores, soils from three cores showed wider ratios indicative for the dominance of plant-derived OM, while the fourth core had narrower ratios, pointing to a larger amount of microbial-derived OM. Generally, C/N ratios decrease with ongoing decomposition (Kramer et al., 2003) as the share of microbially derived OM with its characteristically low C:N ratio increases after depolymerization of plant-derived organic macromolecules, with microbial residues largely binding to mineral particle

surfaces and thereby being stabilized (Connin et al., 2001; Vitousek et al., 2002).

**4.2 POM fractions dominate the C stock stronger than the N stock**

The large POM fractions (fPOM, oPOM) clearly dominated the C stocks (~ 17 kg m$^{-3}$) in the analyzed Cryosols, whereas small POM (oPOMs) and clay-sized MAOM represented slightly more than one-third of the stored C (~ 10 kg m$^{-3}$). This nicely illustrates that rather large plant-derived fragments (see Fig. 9) dominate the C storage in these OM-rich Cryosols. Especially

fPOM, mainly consisting of less decomposed plant material, largely contributes to both C and N stocks. However, in contrast to the C stocks, the oPOMs and clay-sized MAOM fractions act besides fPOM as major contributors to the N stock. A probably accelerated degradation of the fPOM fractions under continued warming would terminate the major contribution to the C stock and, at the same time, release vast amounts of N, which could further foster microbial OM mineralization, thereby increasing the importance of mineral N cycling such as microbial ammonification-immobilization turnover, compared to organic N

cycling. As permafrost-affected soils are often waterlogged during the thawing season with changing oxygen availability and anoxic soil microsites, it can be assumed that in these soils nitrification and denitrification accelerate as well, thereby leading to associated increases in nitrous oxide emissions (Marushchak et al., 2011; Voigt et al., 2017). While a shift from aerobic to anaerobic conditions can hamper the overall decomposition of organic compounds, a shift from anaerobic to aerobic conditions, e.g., when a thawed Arctic soil is exposed to drying conditions, can accelerate decomposition (Keiluweit et al.,

2017). With regard to consequences for the role of plants for C and N budgets, some studies point to more plant available N leading to a changing flora and an increasing plant biomass that will be able to counteract the soil C loss caused by thawing (Sistla et al., 2013; Keuper et al., 2017), while others question that gains in biomass will lead to a sufficient compensation of the loss in soil C (Salmon et al., 2016). No matter which of the predictions proves true, as the rather labile fPOM fractions store about almost one third of the N in these soils, thawing will lead to a profound change in N budget and N cycling with

presumably increasing N bioavailability and increasing importance of mineral N cycling (Voigt et al., 2017; Altshuler et al., 2019).

The C and N content, C/N ratio and decomposition proxies clearly group the particulate OM fractions into large POM (fPOM and oPOM) and small POM (oPOMs). While the large POM fractions showed rather high C/N ratios, the C/N ratios of oPOMs were considerably lower (Table 1, Fig 3). This demonstrates that oPOMs represent a discrete type of SOM consisting of

smaller, more degraded organic fragments intimately connected with mineral particles, a presumption already made by Wagai et al. (2009). We assume that the distinct fibrous structure of the larger POM fractions (see Fig. 9) drives the differentiation into large plant-derived initial POM and mostly microbial dominated small POM in the studied permafrost-affected soils. The fibers act as a hot spot for microbial decay of the larger plant structures (fPOM and oPOM), while providing a specific network that entraps smaller POM and MAOM particles, thereby retaining especially the small POM fraction. The small OM particles

(oPOMs) act as a linking element between the fresh less decomposed plant residues (fPOM, oPOM) and the clay-sized MAOM, to our knowledge a phenomenon not described before in permafrost-affected soils.

**4.3 Molecular and isotopic analyses confirm the differences in the nature of large POM and oPOMs and clay-sized MAOM**

During the decomposition of plant-derived material, the changes in δ$^{13}$C values are usually subtle and are determined by a

variety of factors, especially by the composition of the original plant material (Ågren et al., 1996). Nevertheless, SOM





compounds rich in presumably more recalcitrant macromolecules, like lignin or aromatic hydrocarbons, have lower $\delta^{13}C$ values than labile compounds, like carbohydrates (Schmidt and Gleixner, 1997). Soil $\delta^{13}C$ values depend on several factors, besides the $\delta^{13}C$ value of the plant litter, climatic factors, soil texture and soil processes are decisive (Nel et al., 2018). Here, we found clear positive correlations between the decomposition stage (a-oa ratio, 70-75/52-57 ratio, C/N ratio) of the large POM

fractions and $\delta^{13}C$ (Fig. 8). A negative correlation between $\delta^{13}C$ and rather recalcitrant lipids both in the large POM and oPOMs fractions was demonstrated (Fig. 8), which nicely reflects the relative increase in aliphatic compounds with progressing decomposition (Benner et al., 1987). Although we demonstrate clear mechanistic differences between large (fPOM, oPOM) and small POM (oPOMs) with respect to C sequestration, the intrinsic decomposition in both OM pools follows the same principles. This is also supported by the positive correlation of the $\delta^{13}C$ in large and small POM with the bulk soil C/N ratios.

Thus, the overall elemental composition of the bulk soils can directly be linked to the $^{13}C$ isotopic composition of the fresh and more decomposed POM fractions.

In Arctic ecosystems, $N_2$ fixation is known as the major N input into ecosystems (Granhall and Selander, 1973; Rousk et al., 2017, 2018) with N fixation rates between 1 and 29 kg N ha$^{-1}$ a$^{-1}$, depending on which $N_2$ fixing species (e.g. free-living or moss-associated cyanobacteria) is dominating (Rousk et al., 2017). Furthermore, Arctic soils are known to be dominated by

organic N cycling rather than mineral N cycling (Hobbie and Hobbie, 2008), while atmospheric N deposition is low in this region (Hole et al., 2009). The soil $\delta^{15}N$ values we found are consistent with $\delta^{15}N$ values reported for bacterial $N_2$ fixation as N source (Casciotti, 2009; Hoefs, 2015), but also similar to values reported for plant litter-derived OM (Connin et al., 2001). Other studies reported stable or increasing $\delta^{15}N$ values with advancing decomposition (e.g. Ågren et al., 1996; Connin et al., 2001). Whether increases in $\delta^{15}N$ occur with enhanced decomposition and N turnover is largely depending on gaseous N loss

processes, such as ammonia volatilization, and nitrous oxide and dinitrogen losses through nitrification and denitrification, as highest isotope fractionation factors are reported for these processes, enriching the heavier $^{15}N$ isotope in soil, while $^{14}N$ is preferably lost to the atmosphere (Bedard-Haughn et al., 2003; Nel et al., 2018). By illustrating decreasing $\delta^{15}N$ with increasing OM decomposition, our results seem to contradict this presumption. Therefore, we assume that biological $N_2$ fixation is a decisive control of $\delta^{15}N$ in the studied soils, as also recently shown for permafrost-affected soils of Tibet (Chang et al., 2017).

Such a dominant role of biological $N_2$ fixation in regulating $\delta^{15}N$ requires that nitrification/denitrification and associated gaseous N losses as well as atmospheric inputs are not significant for the studied soils, which is in general agreement with the N cycle paradigm for the High Arctic (Schimel and Bennett, 2004).

By using NMR spectroscopy, we were able to differentiate further between large POM fractions (fPOM, oPOM) and oPOMs and clay-sized MAOM, which also allowed a nice clustering of these materials into distinctly different OM pools with respect

to assumed bioavailability (see the representative example in Fig. S1). The NMR spectra of both large POM fractions were clearly dominated by a major peak around 70 ppm and a minor peak around 105 ppm, both relating to polysaccharides (Koelbl and Koegel-Knabner, 2004). This was well reflected by the calculated high amounts of carbohydrates, the high 70-75/52-57 ratio and low a-oa ratios, which all point to the rather labile undecomposed nature of the larger OM particles.

Based on the combined indications of the decomposition proxies, all pointing in the same direction, we assume a high potential

bioavailability for both large POM fractions (fPOM, oPOM). Interestingly, when comparing the decomposition proxies for these POM fractions per single soil layer, they indicate a less pronounced decomposition for oPOM in most of the samples. These patterns deviate from what is commonly observed in temperate soils, i.e. an increased degree of decomposition with increasing aggregate occlusion and decreasing POM size (fPOM<oPOM<oPOMs) (Mueller and Koegel-Knabner, 2009). We assume that this demonstrates a reduced bioaccessibility (accessibility of OM by microorganisms and enzymes) of oPOM,

which is encrusted by mineral particles, leading to a reduced degree of decomposition of the occluded as compared to the free POM. Thus, the initial microbial decomposition of the surfaces of fresh plant residues (fPOM) driven by microbial decay leads, in part, to the formation of oPOM due to the association with minerals glued to the POM surfaces by microbial residues, e.g. extracellular polymeric substances (Tisdall and Oades, 1982; Schimel and Schaeffer, 2012; Costa et al., 2018). Here, we



demonstrate the soil structure formation in Cryosols as driven by microbial activity at POM surfaces leading to the stabilization
of rather labile POM without necessarily leading to OM with high degrees of decomposition.

In contrast to the large POM fractions, the NMR spectra of oPOMs and clay-sized MAOM were dominated by peaks around
30 ppm representing long-chain structured aliphatic C derived for example from macromolecules like cutin or suberin (Koegel-
Knabner, 2002). However, the dominating group of compounds as calculated by the MMM were carbohydrates for both
fractions (Table 3). Both fractions also showed distinct peaks around 170 to 175 ppm, representing partly esterified carboxyl
groups and amide C that stems predominantly from proteins (Koelbl and Koegel-Knabner, 2004). The clay-sized MAOM
showed distinctly higher amounts of protein C (Table 3) compared to all POM fractions, which corroborates the preferential
association of N-rich microbial residues at mineral surfaces (Kleber et al., 2007; Kopittke et al., 2018, 2020). This highlights
the fact that the association of OM with mineral surfaces presumably follows the same mechanisms as previously described
for temperate soils (Kleber et al., 2007). In the specific context of the studied permafrost-affected soils, the oPOMs represented
some kind of passage fraction. Although it clusters with the clay-sized MAOM in the PCA (Fig. 5), the small POM links to
the large POM fractions as illustrated in Fig. 7. Thus, in contrast to the larger, relatively undecomposed plant residues, lipids
and proteins contribute noteworthy to the oPOMs fractions and the fine MAOM of the clay-sized fraction. This clearly points
to the increased amount of microbial-derived compounds in these fractions, as already stated above with respect to the C/N
ratio and $\delta^{15}N$. Thus, the MAOM in these soils is dominated by microbial-derived SOM rich in biologically fixed N. As shown
in Fig. 9, oPOMs is represented by degraded plant residues, fungal hyphae and amorphous material which can be assumed to
mainly represent microbial necromass (Miltner et al., 2012). The PCA demonstrated that oPOMs represents a linking fraction
between the initial plant residues of the larger POM fractions and the microbial OM dominated clay-sized MAOM (Fig. 5).
However, it falls short to assume that all OM in POM fractions is dynamic and all MAOM is slow-cycling (Torn et al., 2013).
Our results underline that the large and rather undecomposed POM fractions rich in carbohydrates might act as a highly
bioavailable substrate in a warmer future. This means when active layers deepen and the larger POM fractions become
accessible to microorganisms, oPOMs and clay-sized MAOM may represent a C pool that is less bioavailable and thus
presumably more stable. Besides the demonstrated occlusion of particulate OM, we were able to show the quantitative
importance of MAOM for the C storage in these High Arctic soils. Thus, the oPOMs and clay-sized MAOM represent altered
and microbially transformed OM pools that could gain influence regarding C storage under further thawing conditions in soils
of this region. Besides the importance for C sequestration, the high amount of biologically fixed N of the MAOM may also be
released and foster the microbial decay of the high amounts of C stored in larger POM fractions (Jilling et al., 2018).

## 5. Conclusions

Employing physical fractionations and molecular level analyses, we show that the SOM fractions that contribute with about
17 kg C m$^{-3}$ for more than 60 % of the C stocks in the investigated Arctic soils are presumably highly labile and vulnerable to
environmental changes. In the face of global warming, most of this labile C, currently protected from decomposition by cold
temperatures, will be prone to mineralization, with severe consequences for the C stocks in Arctic soils. Thus, relatively stable,
small occluded POM, found to be acting as a link between the larger POM fractions and mineral-associated OM within the
SOM continuum, and clay-sized MAOM that account with 10 kg C m$^{-3}$ for about 40 % of the C stock currently, will likely be
decisive for the quantity of C protected from mineralization in Arctic soils in a warmer future. Using $\delta^{15}N$ as proxy for N
balances indicated an important role of N inputs by biological nitrogen fixation especially for more decomposed organic matter,
while gaseous N losses appear to be of minor importance. This could however change in future, as with about 0.4 kg N m$^{-3}$
one third of the N is present in presumably bioaccessible SOM fractions, which could lead to increases in mineral N cycling
and associated N losses under the auspices of global warming.



*Data availability.* The data that support the findings of this study are available from the corresponding author upon request.

*Supplement.* Supplementary material is available.

*Author contributions.* IP conducted analyses in the laboratory (elemental analysis, NMR measurements) and wrote the
manuscript. SZ was responsible for the sampling and the selection of the respective cores. LCZF conducted analyses in the laboratory (fractionation, elemental analysis, NMR measurements). FB conducted stable isotope measurements. CWM developed the design of the study. IP, MD, GA and CWM were responsible for data evaluation and the interpretation of results. All authors discussed the data and results and contributed to the final form of the manuscript.

*Competing interests.* The authors declare that they have no conflict of interest.

*Acknowledgements.* We thank Maria Greiner for her assistance with the physical soil fractionation and the elemental analysis and Theresa Hautzinger for her support in the laboratory.

*Financial support.* This study was supported through the Cluster of Excellence "CliSAP" (EXC177), University of Hamburg, funded through the German Research Foundation (DFG) and the BMBF project CARBOPERM (03G0836A). The analyses were partly supported by the DFG in the framework of the priority programme 1158 'Antarctic Research with Comparative Investigations in Arctic Ice Areas' (MU 3021/8). The work of MD was supported through the DFG NIFROCLIM project (DA1217/4-1).

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



**Table 1: Mean values of SOM fractions:** C and N stocks (projected to 1 m soil depth) and C/N ratios.

| SOM fraction | C stock kg C m⁻³ | N stock kg N m⁻³ | C/N ratio |
|---|---|---|---|
| fPOM | 14.0±4.6 | 0.3±0.1 | 46±16 |
| oPOM | 2.6±1.1 | 0.1±0.1 | 51±22 |
| oPOMs | 5.4±3.5 | 0.3±0.2 | 17±5 |
| clay-sized MAOM | 4.6±2.2 | 0.4±0.2 | 13±1 |
| silt-sized MAOM | 0.7±0.4 | 0.1±0.0 | 10±1 |
| sand-sized MAOM | 0.2±0.1 | 0.0±0.0 | 10±3 |
| **sum** | **27.5±11.9** | **1.2±0.6** | |



**Table 2: Results of ¹³C CP-MAS NMR spectroscopy:** relative chemical composition of SOM fractions and decomposition proxies (a/o-a ratio and 70-75/52-57 ratio).

| SOM fraction | relative chemical composition[1] | | | | a/o-a ratio[2] | 70-75/52-57 ratio[3] |
|---|---|---|---|---|---|---|
| | alkyl C | O/N alkyl C | aromatic C | carboxyl C | | |
| | ---------------------------- % ---------------------------- | | | | | |
| fPOM | 13.3±5.0 | 70.2±7.6 | 11.6±2.5 | 4.9±1.9 | 0.2±0.1 | 5.6±2.1 |
| oPOM | 12.5±6.0 | 68.5±8.4 | 12.2±4.0 | 6.5±2.6 | 0.2±0.1 | 7.4±3.3 |
| oPOMs | 25.2±5.9 | 52.1±6.3 | 14.0±3.0 | 8.5±2.2 | 0.5±0.2 | 2.6±0.3 |
| clay-sized MAOM | 24.0±2.6 | 49.6±3.4 | 15.1±1.8 | 11.2±3.7 | 0.5±0.1 | 2.1±0.3 |

[1] Relative chemical composition determined by integration of the following chemical shift regions: -10 to 45 ppm (alkyl C), 45 to 110 ppm (O/N alkyl C), 110 to 160 ppm (aromatic C) and 160 to 220 ppm (carboxyl C).
[2] Ratio of alkyl C and O/N alkyl C according to Baldock et al. (1997).
[3] Ratio of the chemical shift regions 70 to 75 ppm and 52 to 57 ppm according to Bonanomi et al. (2013).



**Table 3: Results of 13C CP-MAS NMR spectroscopy:** results from the molecular mixing model. Data obtained according to the molecular mixing model established by Baldock et al. (2004) and Nelson and Baldock (2005); 5 component model (-char) with N:C. constrained.

| SOM fraction | molecular mixing model | | | | |
|---|---|---|---|---|---|
| | carbohydrate | protein | lignin | lipid | carbonyl |
| | ------------------------------------------------ % ------------------------------------------------ | | | | |
| fPOM | 61.4±8.0 | 6.4±2.8 | 21.7±3.2 | 8.2±3.9 | 2.3±1.9 |
| oPOM | 60.9±9.9 | 6.4±3.8 | 21.2±5.5 | 7.6±4.8 | 3.9±4.0 |
| oPOMs | 41.9±5.9 | 17.0±3.2 | 21.3±4.3 | 18.9±5.1 | 0.9±2.0 |
| clay-sized MAOM | 41.1±2.9 | 22.6±2.0 | 21.2±3.7 | 13.5±2.8 | 1.6±2.3 |









**Figure 1: Aerial picture of Samoylov Island**: Red crosses indicate sampling sites, identification numbers of cores are given (Boike et al., 2012).

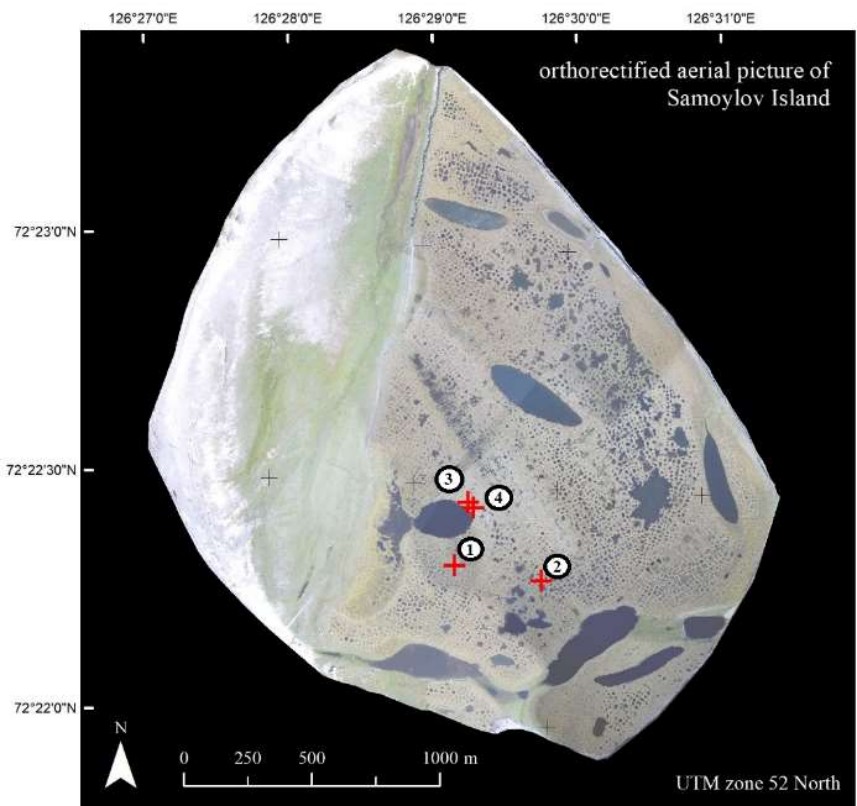










**Figure 2: Content of C (a) and N (b) of bulk soils (I) and SOM fractions (II) (in mg g⁻¹).**

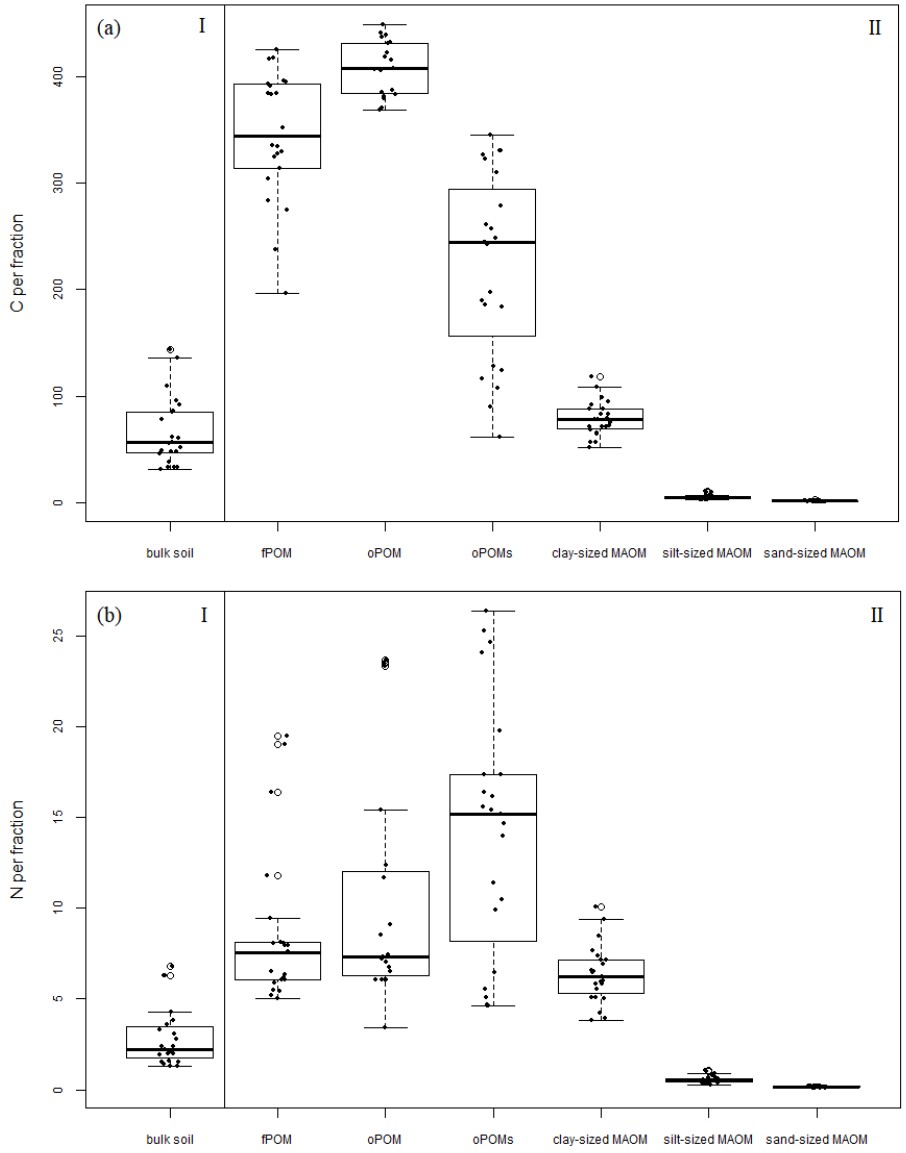





**Figure 3: C/N ratio of bulk soils (a) and SOM fractions (b).**

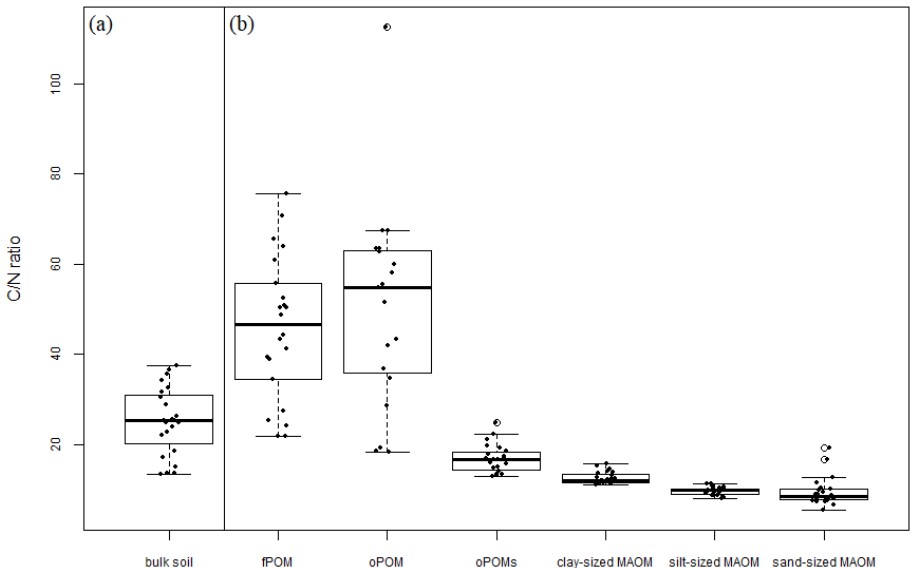














**Figure 4: Stable isotopes plotted against C/N ratios and δ¹⁵N plotted against N content of SOM fractions:** The values of $\delta^{13}C$ (‰ relative to V-PDB) (a) and the values of $\delta^{15}N$ (‰ relative to air $N_2$) (b) in relation to the C/N ratio (log-converted) of SOM fractions and $\delta^{15}N$ (‰ relative to air $N_2$) plotted against N (in mg g⁻¹) content of the SOM fractions (c).



**Figure 5: PCA of the SOM fractions' investigated properties:** PCA of $\delta^{13}C$ (‰ relative to V-PDB) [d13C], $\delta^{15}N$ (‰ relative to air $N_2$) [d15N], C and N content of the fractions [Cfracmgg, Nfracmgg], C/N ratio of fractions [CNfrac] and of bulk soils [CNbulk] and $^{13}C$ CP-MAS NMR-derived decomposition proxies (a/o-a ratio [aoa.ratio] and 70-75/52-57 ratio [bonanomi]).

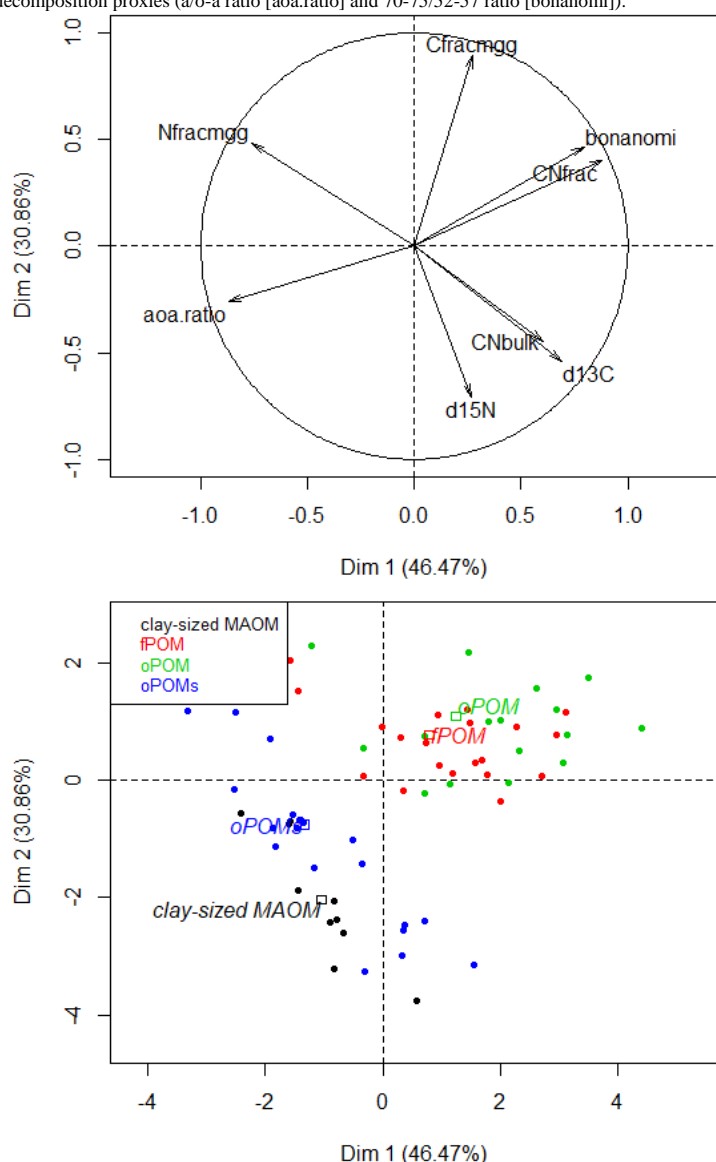



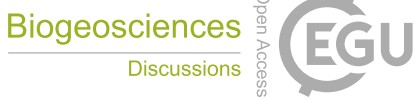

**Figure 6: Decomposition proxies obtained by ¹³C CP-MAS NMR:** Both a/o-a ratio (a) and 70-75/52-57 ratio (b) of SOM fractions demonstrate the similarity of large POM fractions (fPOM and oPOM) and the conjunctive characteristics of the oPOMs fractions that links large POM fractions and the clay-sized MAOM fraction.

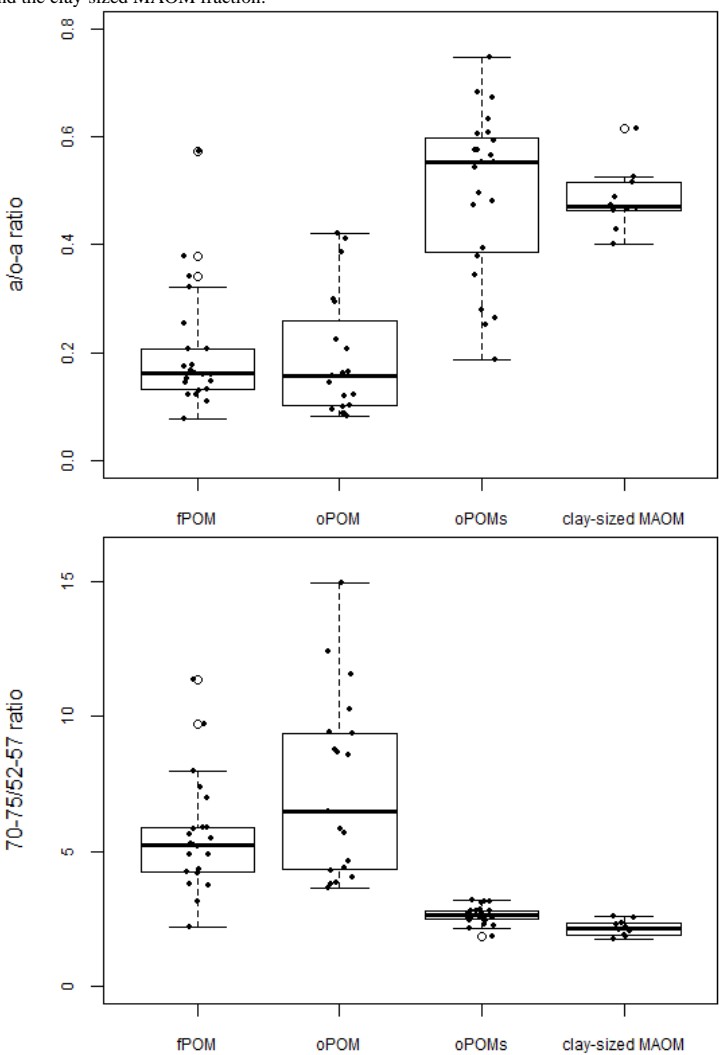








**Figure 7: Relation between decomposition proxies and C/N ratio:** $^{13}$C CP-MAS NMR-derived decomposition proxies a/o-a ratio vs. C/N ratio (a) and 70-75/52-57 ratio vs. C/N ratio (b).

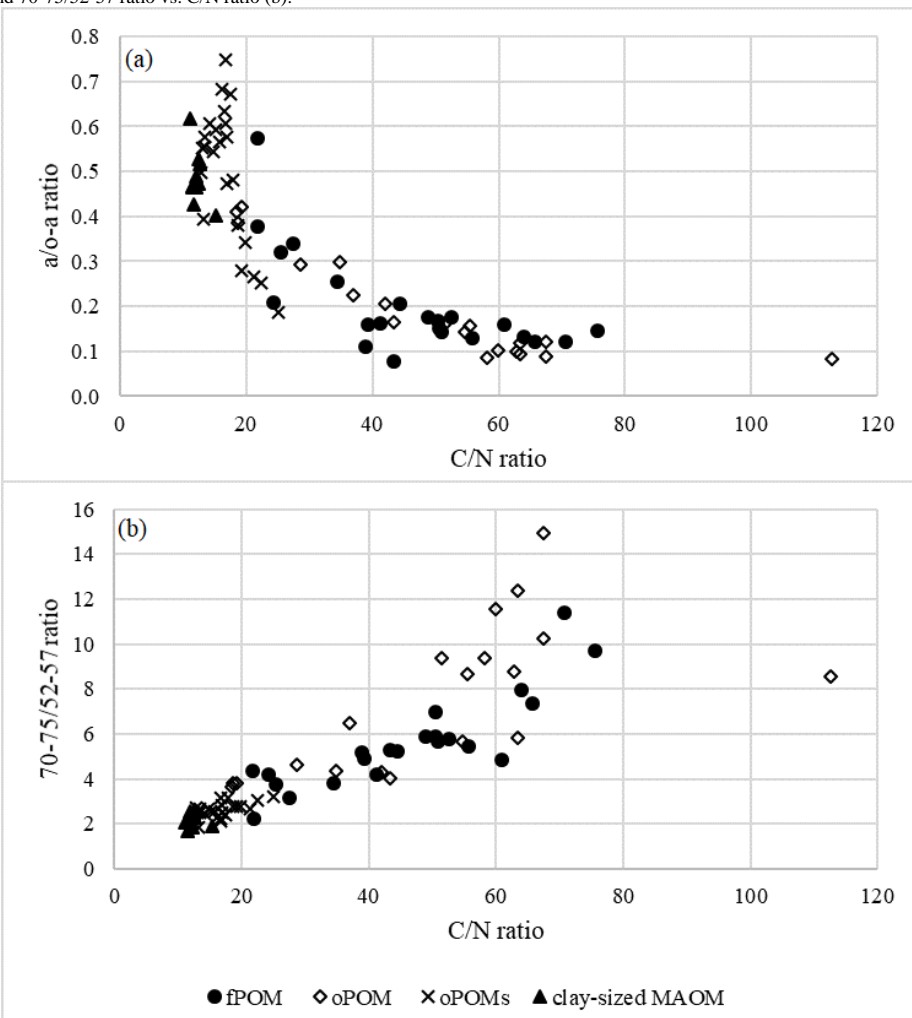










**Figure 8: PCA and correlation matrices of POM fractions:** The large POM (oPOM and fPOM) fractions (a) show different correlations compared to oPOMs fractions (b).

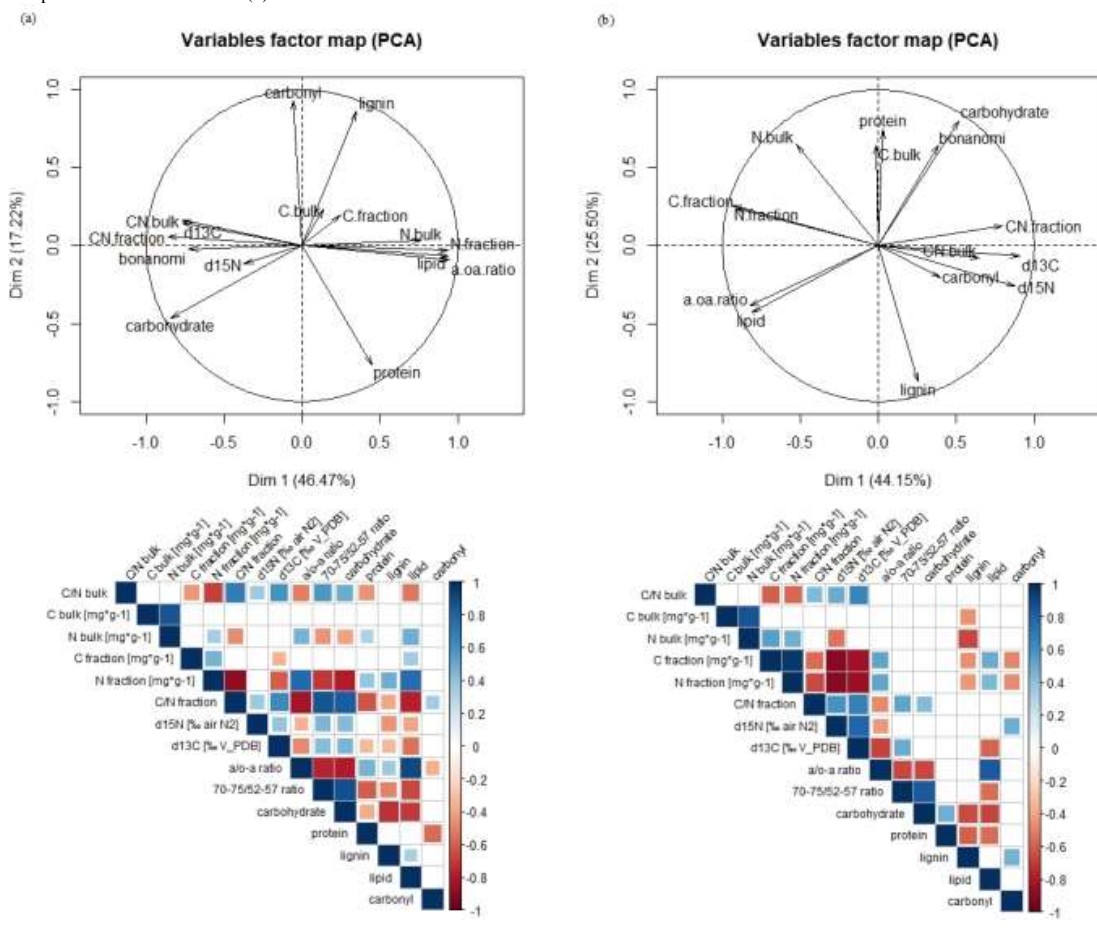








**Figure 9: Differences in POM fractions in SEM imagery:** While fPOM fraction (a) and oPOM fraction (b) consist mainly of parts of plant-derived litter after initial decomposition, the SEM image of the oPOMs fraction (c) of the same sample reveals that mineral and organic
particles are enmeshed tightly.

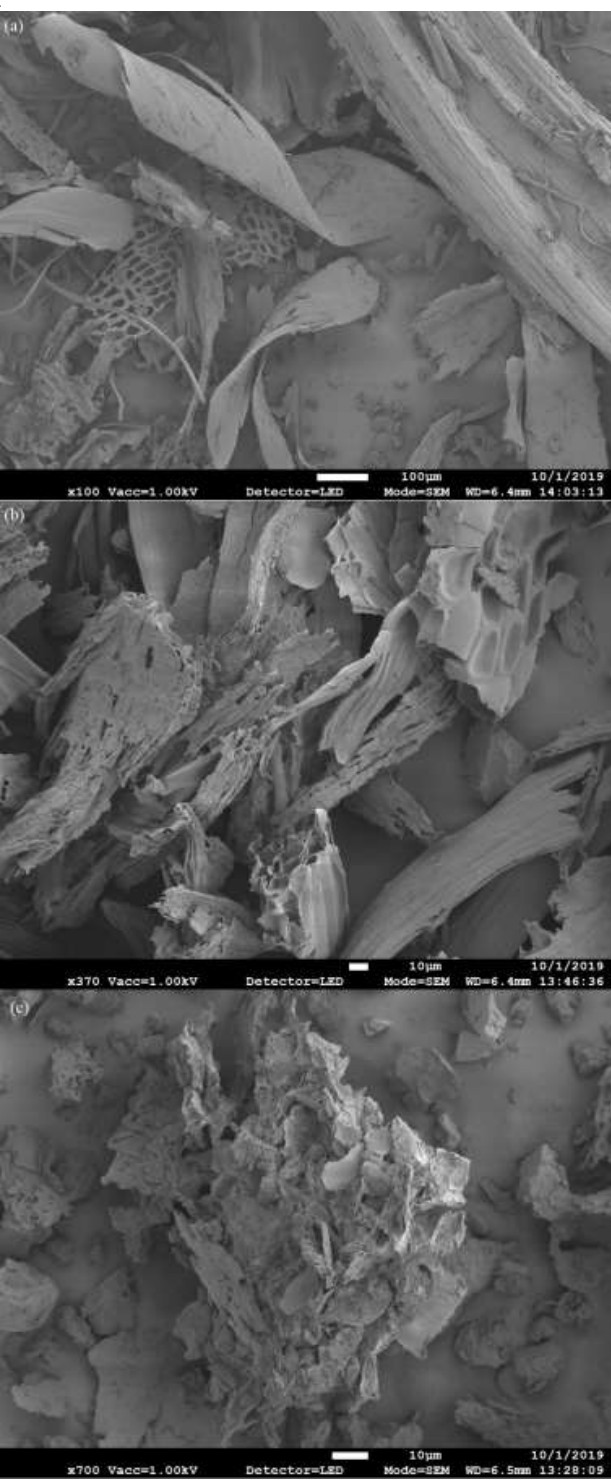