# Peer review of "From fibrous plant residues to mineral-associated organic carbon –"

_Biogeosciences, 2020_

## Referee Comment (RC1) · Anonymous Referee #1 · 19 Mar 2020

The present manuscript by Prater et al. reports new data on various physical fractions of soil organic matter (aggregated, small occulted, mineral associated) in Arctic permafrost-affected environments in the Lena River Delta, Siberia. Several chemical analyses (C and N content, d13C, 13C NMR) were jointly used on the different fractions to better understand the fate of fibrous plant residues in permafrost soils. The manuscript is well written and the presented results and discussion are important for understanding the soil organic matter fate in changing Arctic regions. The authors should consider some minor comments before accepting the manuscript for publication with Biogeosciences.

Specific minor comments and suggestions:

- line 45: please check the reference (should be Frank et al., 2012)

- lines 58-59: the sentence should come before, the warming climate is already mentioned lines 38-39 for example

- lines 75-78: please clarify your objectives, the "physical fractionation" is an approach and not an objective

- line 81: add "Siberia" somewhere

- line 98: add "electric conductivity (EC)" to be consistent with line 117

- line 162: "to detect correlation": check the sentence, statistically a correlation is quantified and not "detected" using a plot

- line 179: the data could be presented with cumulative area charts for each profile and each element (C and N) to illustrate the proportions of each fraction by depth (in supplement)

- lines 221-222: move to the discussion

- line 271: "considerable amount of N" compared to what?

- line 283: change "C:N" to "C/N"

- line 340: change "dinitrogen" to "N2"

- figure 3: use "I" and "II" instead "a" and "b" to be consistent with the figure 2

- figure 4: the quality is too bad, be consistent with fig. 2-3, use indication of x log-scale (and not just 10 and 100 that are not indicative, add minor gridlines for example), use the same color code as in figure 5

- figure 5: change the labels in the PCA to be in agreement with the text (C/N, a/o a ratio, etc.)

- figure 7: same comments as for the Excel plot in figure 4

- figure 8: I do not understand the point of adding both PCA and correlation matrices. I suggest to keep either the PCA, including individuals (as done in figure 5), or correlation matrices only

---

## Referee Comment (RC2) · Anonymous Referee #2 · 20 Mar 2020

The manuscript entitled "From fibrous plant residues to mineral-associated organic carbon – the fate of organic matter in Arctic permafrost soils" analysed four Cryosol soil cores for quantity and quality of organic matter to understand stabilization mechanisms and mineralization potential under climate change. The manuscript is very well written, within the scope of Biogeosciences and informative. No major methodological flaws were detected. The novelty of the results is somewhat limited, the results are close to what should have been expected. However, every new dataset on soils from these remote land masses that are warming up rapidly is valuable per se and the mix of methods is strong, of course. My most important major concern is that I was not able to understand how the authors exactly did the 1 m stock extrapolation when only some

selected layers were fractionated. There is no information on that specific issue and looking at the reference publication reveals that soil profiles (usual for river terraces) were extremely heterogeneous with depth. This depth dependency and spatial heterogeneity in general, is of major importance for upscaling, which is again of specific importance for that vast and SOC rich region. This starts from how single soil profiles are averaged which should thus be done and described with care. A second major issue is related to the data shown in Fig. 4a: How do the authors explain this positive correlation of d13C and C/N ratio within the fractions, I think the opposite would be expected (the more decomposed, the more positive d13C)? Still, my recommendation is publish after some revisions. Please find specific comments below:

General: You might want to consider to call oPOM oPOMl (l for large), this would be more consistent when you also have oPOMs as individual fraction.

l.16: "the permafrost region" appears too unspecific. Is it the Northern Circumpolar Permafrost region or do you include the high mountain ranges here?

l.74: Why Ping et al. at the end?

l.94-97: Please be more specific about the sapling locations, or the selection of these 4 cores. Why those, any criteria to ensure that they are representative for the most likely very heterogeneous area? Can't really find information on that in the cited literature. Also, please mention the soil types. In Zubrzycki et al.2013, different soil types are mentioned. Was it all cryoturbated soils?

l.102: Which selected layers, how many samples were fractionated?

l.102: Table 1 is missing, but would be extremely important, also to judge about some interpretations of the authors.

l.117: During the washing… → How did you separate MAOM and oPOMs? Another density step? This is not clear.

l.126: Projected to 1m: How was this done exactly, and why is there no information

about the depth distribution of fractions if the authors state that different depths were analysed? This is a bit confusing.

l.164: find correlations with what?

l.188: give ranges for the silt and sand-sized MAOM as well, otherwise the sentence reads incomplete.

l.217: This is something that is not clear to me: How do you explain this trend? Usually it should be the other way around. Why is d13C more negative when C/N ratios are decreasing?

Results in general: I was missing a depth distribution of the fractions, all results are depth independent and it is unclear, how homogeneous the profiles were.

l.268: It is not clear to me, if really cryoturbation caused the depth distribution of POM, but again this can only be judged if some depth information is included. Especially in river terraces, it could also be successive growth of the soil profile via sedimentation, and potentially even growth of organic layers.

l.286: Title sounds like POM fractions do also dominate N stocks, which doesn't seem to be the case (table 1). Maybe consider to rephrase.

l.293: "release vasts amount of N" → This is a contradiction to what has been said before and also the title (not much N in POM fractions)

l.313: Where does this information come from that fibres act as hot spots for microbial decay?

l.356: Where is this comparison per single soil layers?

Fig.4: X-axis: Why did you put the x-axis on top of the graph and show only 10 and 100? Is there any reason for that. Readability would improve when numbers are at the bottom and more continuous.

[Figure]

---

## Short Comment (SC1) · 3 Apr 2020

*A note upfront from the submitting person: This review was prepared by four master students in geography at the University of Zurich. The review was part of an exercise during a second semester master level seminar on "the biogeochemistry of plant-soil systems in a changing world", which is organized by prof. Dr. Michael Schmidt and myself. We would like to highlight that the depth of scientific knowledge and technical understanding of these reviewers represents that of master students. We enjoyed discussing the manuscript in the seminar, and hope that the comments will be helpful for the authors.*

[Figure]

The manuscript by Prater et al. provides new data on the different physical fractions of soil organic matter from the Lena River Delta in the Arctic. The area on Samoylov Island is characterized by permafrost. The authors investigated soils with respect to the composition and distribution of organic C among differently stabilized SOM fractions, in order to gain knowledge on the mechanisms stabilizing organic C in Arctic soils, besides impaired decomposition due to low temperatures. The methods consists of the use of sophisticated approaches, separating SOM into different fractions, allowing for a detailed understanding of the stabilization mechanisms of organic carbon in soils. The study is relevant, as there are still rather few analytical approaches to the stabilization mechanisms to assess the variability of C stocks in tundra soils. The research question is particularly relevant as the study deals with a region where permafrost occurs and where soils are both an important store of carbon and other greenhouse gases and are affected by global warming. The authors did not formulate a clear hypothesis or the expected results. Consequently, they did not make a comparison between their results and their original expectations, which makes it difficult to compare the results of the research with other studies.

General comments and suggestions:

- The research question addressed by the authors is important due to the lack of knowledge on the topic. Indeed, researchers only recently started to understand the importance of cold soils for the global carbon cycle, and thus global climate. As a consequence, only a few studies related to this topic have been made so far. - The authors did not explicitly state any hypotheses. They described their intent of investigating the effect of climate change on the carbon stabilization in permafrost-affected soils, but they remained vague and did not state any kind of expected results. Therefore, it is difficult to understand to what extent the research contributed to their question. - The study site is situated in the river delta of the River Lena. Chemical composition and structure of the soil could be the result of flooding which is not the case for typical arctic permafrost soils. In general, the isle may be more affected by the Lena itself

than by the rising temperature. In addition, the closeness of the Siberian sea will have an influence of the isle too, as the ocean moderates the temperatures. Therefore, the study site on the isle Samoylov maybe not representative for arctic permafrost soils in general. - Do you think is the d15N a suitable method? There are many uncertainties related to it, which could be elaborated upon.

Specific comments and suggestions:

L. 75-78: Here the authors write about their approach and the aims, which are basically to gain better knowledge on the topic. Since this section is at the end of the introduction, we think that this part is the most suited for adding the research questions and hypotheses. We think this is important, especially because the authors took four soil cores in a vast area that might be highly heterogeneous. Therefore, having expectations related to the SOM fractions you expect to find in this area, including also the stratification of the soil layers could help determining how representative the four soil cores are with respect to the whole study area.

L. 94-97: In this part the methods are described. However, the authors then state that "a detailed description of the study area and the sampling of the soil cores can be found in Zubrzycki et al. (2013)".We advise that the authors include all relevant information also in the presented manuscript. Otherwise, the readers have to go into the literature to find this relevant information.

L. 101-105: Here, the authors describe how samples were collected, but omitted to state how many samples were collected for each SOM fraction and from which soil core they were collected. We advise to provide the number of samples of each SOM fractionation type, because otherwise it might be difficult to interpret the graphs. We also checked the literature but found any information about the number of samples in Zubrzycki et al., 2013.

Fig. 4: The three graphics (figure 4. a, b, c) could be made more similar. Further, for what concerns figure a and b, the authors represented only the two extreme values

on the x-axis (10 and 100), which makes it difficult to infer the values of the dots in the middle of the graph. Please include more labels on the x-axis to make it more continuous and improve readability. We would also advice to put the x-axis on the bottom for both graphs (a and b) and not once on the top and once on the bottom. Further, we noticed a clear positive correlation between the C/N ratio and the d13C, however, since the C/N ratio usually decreases during ongoing decomposition, we were expecting the opposite trend. We therefore advise to further explain the meaning of this positive correlation.

Fig.7 & 8: Graphs 7 and 8 are difficult to interpret and would require more information in the captions to make the graphs understandable without the reader having to look up more information in the main text.

Title: Why was the word "fibrous" included in the title? Almost all plants residues are fibrous, except for plant exudates. Do you specifically looked at fibrous plant residues omitting exudates? Further, the fate of organic matter sounds somewhat dramatic. We think the title could be shortened to, for example: "From plant residues to mineral-associated organic carbon in Arctic permafrost soils".

---

## Referee Comment (RC3) · Anonymous Referee #3 · 22 Apr 2020

This is a very well-written manuscript that describes organic matter content and composition of physically-isolated density and particle size fractions collected from ice-wedge polygon centers in the Arctic. The objective of the paper is to characterize degree of decomposition of organic matter in permafrost soils with varying degrees of association with mineral surfaces to better understand potential bioavailability of this organic matter pool to warming and thawing. The authors present a thorough chemical characterization of particulate and mineral associated organic matter pools through C and N elemental analysis, stable isotopes and C13-NMR spectroscopy. The results interestingly reveal large contributions of potentially chemically bioavailable POM to the bulk soil C pool, whereas mineral-associated fractions contribute more to the soil N pool.

This work has implications for predictions of the response of similar permafrost-affected soils to warming.

Abstract:

L. 25: "We demonstrate that" It would be helpful in this sentence to operationally identify the fraction being discussed (that is, how was it isolated physically?) to better understand how it is being interpreted as "bioaccessible." Can you define the term bioaccessible? Is it synonymous with the more common "bioavailable" or does it specifically refer to physical accessibility?

Methods: The methods indicate soil drill cores are taken but do not highlight what depths are analyzed and presented. The text states in L. 102 : "Our analyses focused on selected layers only, as shown in Table 1" but Table 1 does not include this information. One would expect that the contribution of POM vs MAOM and the state of decomposition may vary with soil depth (perhaps not in the traditional predictions) yet the paper does not describe what depths are being analyzed.

Discussion:

The discussion is quite long with extensive paragraphs that have multiple ideas, which makes it sometimes a little difficult to follow all the ideas. Consider where the discussion can be streamlined and how paragraphs could be split into smaller blocks of text.

Section 4.1- The section heading is perhaps not the most informative of the text, as permafrost processes (other than one mention to cryoturbation) are not discussed in depth here. Consider renaming the section or including more information on processes. It may also be helpful to separate the text into a paragraph on C and N stocks and another one on composition of SOM, mainly C:N ratios.

Section 4.2: It is very interesting that the POM and MAOM fractions play such different roles in C and N storage in these soils.

Section 4.3: Consider starting the paragraph l. 332 with summarizing results of N dynamics or 15N and their implication as the first sentence on N fixation seems to have no context. This paragraph could also be moved after the NMR paragraph which flows better after the 13C paragraph.

Minor edits:

Introduction, paragraph starting l. 58-78 is too long with too may different ideas. Should be broken up into smaller paragraphs, one on effects of climate change on SOM, one on SOM methods, then the research objectives.

Spell out abbreviations for symbols in the Table legends. For example, fPOM, MAOM. . . Also indicate whether data reported are means and standard error or means and standard deviation.

Table 2. Should a/o-a ratio be O-a ratio? (capital O)

Figure 1. May be helpful to indicate what the white and blue colors are on the image. Ice and open water? Unclear because the ocean is black.

l. 240: add ppm after 70-75 ppm /52-57 ppm

---

## Author Comment (AC1) · 4 May 2020

Dear Referee #1,

We thank you for your kind and helpful comments on our manuscript and we really appreciate that they helped to further improve it. Please find our answers below, we also added the respective line numbers of the updated manuscript to improve the traceability:

- line 45: please check the reference (should be Frank et al., 2012)

Thank you for this hint, the reference reads now Frank et al, 2012 (l. 51).

- lines 58-59: the sentence should come before, the warming climate is already mentioned lines 38-39 for example

As recommended, we moved the sentence (l. 43/44).

- lines 75-78: please clarify your objectives, the "physical fractionation" is an approach and not an objective

We rephrased the sentence to clarify that the detailed insights into chemical composition and stabilization mechanisms of SOM are the objectives and not the fractionation itself (l.80-86).

- line 81: add "Siberia" somewhere

We added "Siberian" (l. 89).

- line 98: add "electric conductivity (EC)" to be consistent with line 117

Thank you for the remark, we added (EC) (l. 109).

- line 162: "to detect correlation": check the sentence, statistically a correlation is quantified and not "detected" using a plot

We changed "detect correlations" to "identify interrelations" (l. 173).

- line 179: the data could be presented with cumulative area charts for each profile and each element (C and N) to illustrate the proportions of each fraction by depth (in supplement)

We thank you for this remark; it is always good to explore better ways to visualize data. We tested the cumulative approach, but decided to stick to the table as we think that a graphical depiction would be redundant.

- lines 221-222: move to the discussion

We moved this part of the sentence to l. 338/339.

- line 271: "considerable amount of N" compared to what?

We changed "considerable" to "noteworthy" (l. 284) as this expression works without comparison. We do not aim at comparing the values; we want to show that the N content should not be neglected.

- line 283: change "C:N" to "C/N"

Thank you for this hint, we changed it accordingly (l. 296).

- line 340: change "dinitrogen" to "N2"

To meet both this recommendation and the recommendations of the co-authors, we changed this part to "nitrous oxide (N2O) and dinitrogen (N2)" (l. 365).

- figure 3: use "I" and "II" instead "a" and "b" to be consistent with the figure 2

We reworked all our figures according to your suggestions and we appreciate that you helped to clearly improve them. We changed the captions and labels accordingly, used a consistent color code for all figures (that should also work for people with color vision deficiency) and took care of a much better quality.

- figure 4: the quality is too bad, be consistent with fig. 2-3, use indication of x log-scale (and not just 10 and 100 that are not indicative, add minor gridlines for example), use the same color code as in figure 5

We changed it accordingly, see comment above (figure 3).

- figure 5: change the labels in the PCA to be in agreement with the text (C/N, a/o a ratio, etc.)

We changed the figure, see comment above (figure 3).

- figure 7: same comments as for the Excel plot in figure 4

We changed the figure accordingly, see comment above (figure 3).
- figure 8: I do not understand the point of adding both PCA and correlation matrices. I suggest to keep either the PCA, including individuals (as done in figure 5), or correlation matrices only.

We decided to follow your suggestion and left only the correlation matrices in the manuscript as we are convinced that those help best to illustrate the differences between large POM fractions and oPOMs fraction.
* * *
[Figure]

orthorectified aerial picture of
Samoylov Island

126°27'0"E    126°28'0"E    126°29'0"E    126°30'0"E    126°31'0"E

72°23'0"N

72°22'30"N

72°22'0"N

N

0    250    500         1000 m

UTM zone 52 north

**Fig. 1.** On this aerial image of Samoylov Island, the separation between the floodplain in the west (with the white unvegetated sandy sediment) and the Holocene terrace on the eastern part (with blue-grey spot

[Figure]

[Figure]

**Fig. 2.** The content of C (a) and N (b) of bulk soils (I) and SOM fractions (free particulate OM (fPOM), occluded particulate OM (oPOM), small occluded particulate OM (oPOMs) and clay-sized mineral-associated

[Figure]

**Fig. 3.** C/N ratios of bulk soils (I) and SOM fractions (free particulate OM (fPOM), occluded particulate OM (oPOM), small occluded particulate OM (oPOMs) and clay-sized mineral-associated OM (MAOM)) (II).

[Figure]

**Fig. 4.** Natural abundance of $\delta$13C and $\delta$15N plotted against the C/N ratios, and the $\delta$15N values plotted against the N content of SOM fractions (free particulate OM (fPOM), occluded particulate OM (oPOM), sma

[Figure]

[Figure]

[Figure]

**Fig. 5.** Principal Component Analysis (PCA) of $\delta$13C (‰ relative to V-PDB), $\delta$15N (‰ relative to air N2), C and N content of the SOM fractions (free particulate OM (fPOM), occluded particulate OM (oPOM), small o

[Figure]

[Figure]

**Fig. 6.** Decomposition proxies obtained by 13C CP-MAS NMR spectroscopy for specific SOM fractions: Both a/o-a ratio (a) and 70-75/52-57 ratio (b) of SOM fractions demonstrate the similarity of large particulat

**Fig. 7.** Relation between decomposition proxies and C/N ratio of distinct SOM fractions: 13C
CP-MAS NMR spectroscopy-derived decomposition proxies a/o-a ratio (a) and 70-75/52-57
ratio (b) vs. C/N ratio for fr

[Figure]

**Fig. 8.** Correlation matrices of POM fractions: The large POM (oPOM and fPOM) fractions (a) show different correlations compared to oPOMs fractions (b). The more intense the color and the smaller the ellipse,

---

## Author Comment (AC2) · 4 May 2020

Dear Referee #2,

We thank you for your valuable and very helpful comments on our manuscript! We appreciate that they supported us substantially with further improving it. Please find our answers to your remarks below, we also added the respective line numbers of the updated manuscript to improve the traceability:

General: You might want to consider to call oPOM oPOMl (l for large), this would be more consistent when you also have oPOMs as individual fraction.

[Figure]

We agree that your suggestion could imply some consistency in the naming. We discussed this and decided to keep our naming as the use of oPOMl and oPOMs could imply the existence of at least one more oPOM fraction between oPOMl and oPOMs and could therefore cause confusion. Furthermore, there are numerous publications that use the same terminology and we would like to maintain comparability for the reader.

l.16: "the permafrost region" appears too unspecific. Is it the Northern Circumpolar Permafrost region or do you include the high mountain ranges here?

We agree that this term is relatively broad. In l. 18 we mentioned the Arctic and according to your suggestion we additionally added "Northern circumpolar" to avoid further misunderstandings (l. 19).

l.74: Why Ping et al. at the end?

We placed Ping et al. 2015 here (l. 79), as we think this paper provides a great overview on the facts that we stated before.

l.94-97: Please be more specific about the sapling locations, or the selection of these 4 cores. Why those, any criteria to ensure that they are representative for the most likely very heterogeneous area? Can't really find information on that in the cited literature. Also, please mention the soil types. In Zubrzycki et al.2013, different soil types are mentioned. Was it all cryoturbated soils?

You are right that this is a very important point in our study. We further specified the sampling area (l. 104), and referred here directly to the Holocene river terrace and also added another reference with more details on the sampling area (l. 108). We also added information on the soils (l. 93-95).

l.102: Which selected layers, how many samples were fractionated? l.102: Table 1 is missing, but would be extremely important, also to judge about some interpretations of the authors.

We thank you very much for this crucial hint. We moved the respective table to the Supplement, but at this point (l.113) we did not change the reference. The information on the samples as the depth layers etc. is now given in table S1. We also added the number of the selected layers we fractionated (l. 113).

l.117: During the washing... → How did you separate MAOM and oPOMs? Another density step? This is not clear.

As this step is very fundamental for our study we are thankful for your remark. To better clarify the procedure, we added more detailed information (l. 128/129). We followed a standard density fractionation approach, where we separated the POM fractions (including oPOMs) from the MAOM by density fractionation, which we describe in l. 116-129. The oPOMs was not separated from the MAOM, but from the oPOM fraction as described in l. 128/129.

l.126: Projected to 1m: How was this done exactly, and why is there no information about the depth distribution of fractions if the authors state that different depths were analysed? This is a bit confusing.

The stocks for the respective sampled and analyzed soil depths were taken as a bulk and projected to one cubic meter. As common practice, we report the stocks based on the sampled and analyzed material. The depth distribution and all according information can be found in table S1.

l.164: find correlations with what?

We rephrased this sentence (l. 173) to better clarify the statement.

l.188: give ranges for the silt and sand-sized MAOM as well, otherwise the sentence reads incomplete.

Thank you, we totally agree and added the missing information (l. 200/201).

l.217: This is something that is not clear to me: How do you explain this trend? Usually

it should be the other way around. Why is d13C more negative when C/N ratios are decreasing?

You are pointing to a very interesting aspect of our work that nicely demonstrates the specificity of the studied soil systems. We discuss this in more detail in the discussion section 4.3 (l. 338-356). We rewrote this section to better emphasize the differences in the 13C abundance for different SOM fractions.

Results in general: I was missing a depth distribution of the fractions, all results are depth independent and it is unclear, how homogeneous the profiles were.

You are right that the depth distribution of fractions is an important fact. We are now providing all according information in table S1.

l.268: It is not clear to me, if really cryoturbation caused the depth distribution of POM, but again this can only be judged if some depth information is included. Especially in river terraces, it could also be successive growth of the soil profile via sedimentation, and potentially even growth of organic layers.

You are right, in areas that are flooded regularly a burial of organic matter can be expected. This was the reason we avoided such areas and did not take samples from the floodplain. We only sampled the terrace that is only very rarely flooded, which is thought not to lead to the burial of pockets of organic matter as the ones discovered. Related information on the depth distribution is given in table S1.

l.286: Title sounds like POM fractions do also dominate N stocks, which doesn't seem to be the case (table 1). Maybe consider to rephrase.

In the studied soils, the particulate OM represents an important N storage pool. While the C stock is dominated by the large POM fractions, the N stock is dominated by the fPOM and the oPOMs fraction – especially the fPOM fraction plays a crucial role for both stocks. We discuss this in l. 302-306.

l.293: "release vasts amount of N" → This is a contradiction to what has been said

before and also the title (not much N in POM fractions)

We found relatively high amounts of N in fPOM, oPOMs and clay-sized MAOM, therefore, we assume an increased release of this N under ongoing warming. With the title we want to indicate that the POM fractions are more important for the C stock than for the N stock, not that the POM fractions are negligible.

l.313: Where does this information come from that fibres act as hot spots for microbial decay?

Thank you for this hint, we added a source for this information (l. 329) and rewrote parts of the respective section. The fibrous OM particles represent relatively undecomposed plant residues, as we were able to demonstrate using NMR spectroscopy. Thus, these particles represent detrital material that most likely provides highly bioavailable OM sources for microbial activity. This hot spot effect of the detritusphere for microbial activity is well known and was addressed in numerous other studies (e.g. Beare et al. 1995, Poll et al. 2006, 2008, Sanaullah et al. 2016).

l.356: Where is this comparison per single soil layers?

Thank you for this hint, we added the respective information (l. 382).

Fig.4: X-axis: Why did you put the x-axis on top of the graph and show only 10 and 100? Is there any reason for that. Readability would improve when numbers are at the bottom and more continuous.

Thank you for this remark, we reworked all of our figures and changed fig. 4 according to your suggestion.

Please also note the supplement to this comment:
https://www.biogeosciences-discuss.net/bg-2020-52/bg-2020-52-AC2-supplement.pdf

**Supplement:**

*Supplement of*

**From fibrous plant residues to mineral-associated organic carbon – the fate of organic matter in Arctic permafrost soils**

**Isabel Prater et al.**

*Correspondence to:* Isabel Prater (i.prater@tum.de)

**Table S1:** Basic properties of bulk soil samples and result of the fractionation: pH (measured in $H_2O$), EC ($\mu S\ cm^{-1}$), bulk density (g $cm^{-3}$), C (mg $g^{-1}$), N (mg $g^{-1}$), C/N ratio and the distribution of SOM fractions (mg $g^{-1}$) for the bulk soil samples of all analyzed depth layers. The particulate organic matter fractions are free (fPOM), occluded (oPOM) and small occluded (oPOMs) particulate organic matter.

| | depth | pH | EC | bulk density | C | N | C/N ratio | distribution of SOM fractions | | | | | |
| | | | | | | | | *particulate OM fractions* | | | | *mineral-associated OM fractions* | |
| | | | | | | | | fPOM | oPOM | oPOMs | clay-sized | silt-sized | sand-sized |
| | cm | (H₂O) | µS cm⁻¹ | g cm⁻³ | mg g⁻¹ | mg g⁻¹ | | | | | mg g⁻¹ | | |
| core 1 | 11-22 | 5.5 | 69 | 0.3 | 55.2 | 1.6 | 36 | 107.6 | 5.9 | 4.4 | 98.4 | 192.3 | 591.5 |
| | 22-30 | 6.5 | 153 | 0.3 | 61.9 | 2.2 | 29 | 103.6 | 16.5 | 9.0 | 141.6 | 294.2 | 435.1 |
| | 30-40 | 6.2 | 87 | 0.4 | 32.9 | 1.4 | 24 | 36.4 | 14.0 | 9.5 | 113.5 | 257.6 | 568.9 |
| | 40-50 | 6.0 | 98 | 0.3 | 48.9 | 1.5 | 33 | 89.7 | 13.4 | 7.4 | 138.3 | 286.5 | 464.6 |
| | 50-62 | 5.4 | 69 | 0.6 | 47.5 | 1.9 | 25 | 70.6 | 12.1 | 12.1 | 134.5 | 243.9 | 527.8 |
| | 62-75 | 5.3 | 87 | 0.4 | 60.3 | 2.4 | 25 | 91.2 | 19.2 | 17.7 | 181.7 | 311.8 | 378.4 |
| core 2 | 30-40 | 5.1 | 117 | 0.4 | 95.8 | 2.8 | 34 | 223.7 | 27.0 | 5.4 | 171.7 | 314.8 | 257.4 |
| | 40-50 | 5.1 | 174 | 0.3 | 109.8 | 3.6 | 31 | 260.6 | 20.8 | 13.1 | 244.5 | 336.8 | 124.2 |
| | 50-60 | 5.1 | 240 | 0.2 | 144.0 | 3.8 | 38 | 295.0 | 22.9 | 218.4 | 122.5 | 247.2 | 94.1 |
| | 60-70 | 4.9 | 203 | 0.4 | 61.7 | 2.4 | 26 | 99.2 | 57.8 | 267.2 | 101.3 | 395.6 | 79.0 |
| core 3 | 11-20 | 6.6 | 85 | 0.5 | 45.8 | 1.3 | 37 | 62.8 | 52.5 | 82.3 | 37.2 | 182.4 | 582.8 |
| | 20-30 | 6.1 | 75 | 0.9 | 31.6 | 1.3 | 25 | 30.2 | 30.0 | 116.0 | 43.7 | 255.1 | 525.0 |
| | 40-50 | 5.8 | 134 | 0.4 | 47.9 | 2.1 | 23 | 169.6 | 20.5 | 176.4 | 97.7 | 305.2 | 230.6 |
| | 59-68 | 6.0 | 76 | 0.9 | 52.2 | 2.0 | 26 | 115.6 | 11.1 | 125.4 | 60.1 | 316.6 | 371.2 |
| | 68-80 | 5.8 | 124 | 0.4 | 135.7 | 4.3 | 32 | 206.8 | 71.7 | 167.1 | 124.8 | 301.6 | 128.1 |
| core 4 | 0-10 | 5.5 | 76 | 0.4 | 56.2 | 3.3 | 17 | 155.5 | 6.6 | 18.2 | 119.6 | 197.0 | 503.0 |
| | 10-20 | 5.8 | 66 | 0.6 | 33.2 | 1.5 | 22 | 75.7 | 3.0 | 3.9 | 121.8 | 241.9 | 553.7 |
| | 20-30 | 5.6 | 82 | 0.6 | 37.6 | 2.0 | 19 | 50.2 | 8.9 | 12.7 | 148.6 | 315.8 | 463.9 |
| | 30-40 | 5.6 | 125 | 0.4 | 77.8 | 3.1 | 25 | 202.0 | 15.2 | 15.8 | 176.8 | 265.5 | 324.7 |
| | 40-50 | 6.0 | 73 | 0.9 | 33.5 | 2.2 | 15 | 16.2 | 8.7 | 22.8 | 163.5 | 366.1 | 422.8 |
| | 50-60 | 5.7 | 121 | 0.7 | 84.5 | 6.3 | 13 | 10.6 | 34.3 | 111.7 | 214.5 | 439.6 | 189.3 |
| | 60-70 | 5.5 | 161 | 0.3 | 85.5 | 6.3 | 14 | 11.3 | 37.2 | 125.5 | 216.7 | 467.0 | 142.2 |
| | 70-79 | 5.6 | 147 | 0.3 | 91.7 | 6.8 | 14 | 13.7 | 30.2 | 133.0 | 231.6 | 479.3 | 112.2 |

**Table S2:** Properties of SOM fractions: C per fraction (mg C (soil g)$^{-1}$), N per fraction (mg N (soil g)$^{-1}$), C/N ratio, $\delta^{13}$C (‰ V-PDB) and $\delta^{15}$N (‰ air N$_2$) for free particulate (fPOM), occluded particulate (oPOM), small occluded particulate (oPOMs) and clay-sized mineral-associated organic matter (MAOM).

[revised manuscript text omitted]

---

## Author Comment (AC4) · 4 May 2020

Dear Referee #3,

We are very grateful for your helpful and supporting comments on our manuscript that help to further improve it. Please find our answers to your remarks below in italics, we also added the respective line numbers of the updated manuscript to improve the traceability:

L. 25: "We demonstrate that" It would be helpful in this sentence to operationally identify the fraction being discussed (that is, how was it isolated physically?) to better understand how it is being interpreted as "bioaccessible." Can you define the term bioaccessible? Is it synonymous with the more common "bioavailable" or does it specifically refer to physical accessibility?

We agree that these terms are often used in a confusing way. When we aim at emphasizing the spatial inaccessibility of the OM, we use "bioaccessibility". When we refer to the microbial availability determined by the chemical composition of a substrate, we use "bioavailability". We made some changes in our manuscript according to our remark.

Methods: The methods indicate soil drill cores are taken but do not highlight what depths are analyzed and presented. The text states in L. 102 : "Our analyses focused on selected layers only, as shown in Table 1" but Table 1 does not include this information. One would expect that the contribution of POM vs MAOM and the state of decomposition may vary with soil depth (perhaps not in the traditional predictions) yet the paper does not describe what depths are being analyzed.

We thank you very much for this crucial hint that we were not aware of. We moved the respective table to the Supplement, but at this point (l. 113) we did not change the reference. The information on the samples like depth layers etc. is now given in table S1 and we corrected the reference accordingly.

Discussion: The discussion is quite long with extensive paragraphs that have multiple ideas, which makes it sometimes a little difficult to follow all the ideas. Consider where the discussion can be streamlined and how paragraphs could be split into smaller blocks of text.

According to your suggestion, we restructured the discussion to increase the readability of the manuscript and split the paragraph discussing stable isotopes and NMR results into two paragraphs.

Section 4.1: The section heading is perhaps not the most informative of the text, as

permafrost processes (other than one mention to cryoturbation) are not discussed in depth here. Consider renaming the section or including more information on processes. It may also be helpful to separate the text into a paragraph on C and N stocks and another one on composition of SOM, mainly C:N ratios.

According to your suggestion we changed the heading to "Cryoturbation determines bulk soil organic matter distribution". As we do not widely discuss the C and N stocks, the aim of the manuscript is clearly on the composition of the SOM fractions, thus we would like to stick to the current paragraph.

Section 4.2: It is very interesting that the POM and MAOM fractions play such different roles in C and N storage in these soils.

We are happy that you acknowledge that this is an interesting finding in our study.

Section 4.3: Consider starting the paragraph l. 332 with summarizing results of N dynamics or 15N and their implication as the first sentence on N fixation seems to have no context. This paragraph could also be moved after the NMR paragraph which flows better after the 13C paragraph.

We followed your suggestion above and separated the paragraph further. We have now one paragraph (4.3) discussing d13C and d15N and we slightly rearranged this paragraph. Another paragraph (4.4) is now only focusing on the NMR discussion.

Minor edits: Introduction, paragraph starting l. 58-78 is too long with too may different ideas. Should be broken up into smaller paragraphs, one on effects of climate change on SOM, one on SOM methods, then the research objectives.

We slightly restructured the Introduction according to your suggestion.

Spell out abbreviations for symbols in the Table legends. For example, fPOM, MAOM... Also indicate whether data reported are means and standard error or means and standard deviation.

[Figure]

Thank you for this remark, we added the missing information to the captions of the tables.

Table 2. Should a/o-a ratio be O-a ratio? (capital O)

This ratio relates to functional groups that consist of O/N-alkyl-C and Alkyl-C, which is normally given as "Alkyl-C to O/N-alkyl-C ratio". To make it easier to read, we defined the ratio of alkyl C to O/N alkyl C as a/o-a ratio in the method section (2.4) and we kept this wording throughout the manuscript and in the tables and figures as well.

Figure 1. May be helpful to indicate what the white and blue colors are on the image. Ice and open water? Unclear because the ocean is black.

Thank you for this important remark, we added the information according to your suggestion. The white color mainly in the western part is the unvegetated sandy sediment of the floodplain and the blue spots indicate water: larger water bodies and shallow water on the terrace. The complete caption reads now "On this aerial image of Samoylov Island, the separation between the floodplain in the west (with the white unvegetated sandy sediment) and the Holocene terrace on the eastern part (with blue-grey spots indicating shallow water and larger water bodies). Red crosses indicate the sampling sites and the identification numbers of the cores are given (Boike et al., 2012)."

l. 240: add ppm after 70-75 ppm /52-57 ppm

We introduce the ratio according to Bonanomi et al. (2013) in the methods section (2.4), where we clearly state the chemical shift regions that are considered for this decomposition proxy. To increase readability we use a reduced naming of the ratio which is in accordance with the "a/o-a ratio" term. Thus, we would like to be consistent in the form that we use to express NMR-derived decomposition proxies.

Please also note the supplement to this comment:
https://www.biogeosciences-discuss.net/bg-2020-52/bg-2020-52-AC4-supplement.pdf

orthorectified aerial picture of
Samoylov Island

UTM zone 52 north

**Fig. 1.** On this aerial image of Samoylov Island, the separation between the floodplain in the west (with the white unvegetated sandy sediment) and the Holocene terrace on the eastern part (with blue-grey spot

**Supplement:**

*Supplement of*

**From fibrous plant residues to mineral-associated organic carbon – the fate of organic matter in Arctic permafrost soils**

**Isabel Prater et al.**

*Correspondence to:* Isabel Prater (i.prater@tum.de)

**Table S1:** Basic properties of bulk soil samples and result of the fractionation: pH (measured in $H_2O$), EC ($\mu S\ cm^{-1}$), bulk density (g $cm^{-3}$), C (mg $g^{-1}$), N (mg $g^{-1}$), C/N ratio and the distribution of SOM fractions (mg $g^{-1}$) for the bulk soil samples of all analyzed depth layers. The particulate organic matter fractions are free (fPOM), occluded (oPOM) and small occluded (oPOMs) particulate organic matter.

| | depth | pH | EC | bulk density | C | N | C/N ratio | distribution of SOM fractions | | | | | |
| | | | | | | | | *particulate OM fractions* | | | | *mineral-associated OM fractions* | |
| | | | | | | | | fPOM | oPOM | oPOMs | clay-sized | silt-sized | sand-sized |
| | cm | (H₂O) | µS cm⁻¹ | g cm⁻³ | mg g⁻¹ | mg g⁻¹ | | | | | mg g⁻¹ | | |
| core 1 | 11-22 | 5.5 | 69 | 0.3 | 55.2 | 1.6 | 36 | 107.6 | 5.9 | 4.4 | 98.4 | 192.3 | 591.5 |
| | 22-30 | 6.5 | 153 | 0.3 | 61.9 | 2.2 | 29 | 103.6 | 16.5 | 9.0 | 141.6 | 294.2 | 435.1 |
| | 30-40 | 6.2 | 87 | 0.4 | 32.9 | 1.4 | 24 | 36.4 | 14.0 | 9.5 | 113.5 | 257.6 | 568.9 |
| | 40-50 | 6.0 | 98 | 0.3 | 48.9 | 1.5 | 33 | 89.7 | 13.4 | 7.4 | 138.3 | 286.5 | 464.6 |
| | 50-62 | 5.4 | 69 | 0.6 | 47.5 | 1.9 | 25 | 70.6 | 12.1 | 12.1 | 134.5 | 243.9 | 527.8 |
| | 62-75 | 5.3 | 87 | 0.4 | 60.3 | 2.4 | 25 | 91.2 | 19.2 | 17.7 | 181.7 | 311.8 | 378.4 |
| core 2 | 30-40 | 5.1 | 117 | 0.4 | 95.8 | 2.8 | 34 | 223.7 | 27.0 | 5.4 | 171.7 | 314.8 | 257.4 |
| | 40-50 | 5.1 | 174 | 0.3 | 109.8 | 3.6 | 31 | 260.6 | 20.8 | 13.1 | 244.5 | 336.8 | 124.2 |
| | 50-60 | 5.1 | 240 | 0.2 | 144.0 | 3.8 | 38 | 295.0 | 22.9 | 218.4 | 122.5 | 247.2 | 94.1 |
| | 60-70 | 4.9 | 203 | 0.4 | 61.7 | 2.4 | 26 | 99.2 | 57.8 | 267.2 | 101.3 | 395.6 | 79.0 |
| core 3 | 11-20 | 6.6 | 85 | 0.5 | 45.8 | 1.3 | 37 | 62.8 | 52.5 | 82.3 | 37.2 | 182.4 | 582.8 |
| | 20-30 | 6.1 | 75 | 0.9 | 31.6 | 1.3 | 25 | 30.2 | 30.0 | 116.0 | 43.7 | 255.1 | 525.0 |
| | 40-50 | 5.8 | 134 | 0.4 | 47.9 | 2.1 | 23 | 169.6 | 20.5 | 176.4 | 97.7 | 305.2 | 230.6 |
| | 59-68 | 6.0 | 76 | 0.9 | 52.2 | 2.0 | 26 | 115.6 | 11.1 | 125.4 | 60.1 | 316.6 | 371.2 |
| | 68-80 | 5.8 | 124 | 0.4 | 135.7 | 4.3 | 32 | 206.8 | 71.7 | 167.1 | 124.8 | 301.6 | 128.1 |
| core 4 | 0-10 | 5.5 | 76 | 0.4 | 56.2 | 3.3 | 17 | 155.5 | 6.6 | 18.2 | 119.6 | 197.0 | 503.0 |
| | 10-20 | 5.8 | 66 | 0.6 | 33.2 | 1.5 | 22 | 75.7 | 3.0 | 3.9 | 121.8 | 241.9 | 553.7 |
| | 20-30 | 5.6 | 82 | 0.6 | 37.6 | 2.0 | 19 | 50.2 | 8.9 | 12.7 | 148.6 | 315.8 | 463.9 |
| | 30-40 | 5.6 | 125 | 0.4 | 77.8 | 3.1 | 25 | 202.0 | 15.2 | 15.8 | 176.8 | 265.5 | 324.7 |
| | 40-50 | 6.0 | 73 | 0.9 | 33.5 | 2.2 | 15 | 16.2 | 8.7 | 22.8 | 163.5 | 366.1 | 422.8 |
| | 50-60 | 5.7 | 121 | 0.7 | 84.5 | 6.3 | 13 | 10.6 | 34.3 | 111.7 | 214.5 | 439.6 | 189.3 |
| | 60-70 | 5.5 | 161 | 0.3 | 85.5 | 6.3 | 14 | 11.3 | 37.2 | 125.5 | 216.7 | 467.0 | 142.2 |
| | 70-79 | 5.6 | 147 | 0.3 | 91.7 | 6.8 | 14 | 13.7 | 30.2 | 133.0 | 231.6 | 479.3 | 112.2 |

**Table S2:** Properties of SOM fractions: C per fraction (mg C (soil g)$^{-1}$), N per fraction (mg N (soil g)$^{-1}$), C/N ratio, $\delta^{13}$C (‰ V-PDB) and $\delta^{15}$N (‰ air N$_2$) for free particulate (fPOM), occluded particulate (oPOM), small occluded particulate (oPOMs) and clay-sized mineral-associated organic matter (MAOM).

[revised manuscript text omitted]

---

## Author Response (AR1)

The present manuscript by Prater et al. reports new data on various physical fractions of soil organic matter (aggregated, small occulted, mineral associated) in Arctic permafrost-affected environments in the Lena River Delta, Siberia. Several chemical analyses (C and N content, d13C, 13C NMR) were jointly used on the different fractions to better understand the fate of fibrous plant residues in permafrost soils. The manuscript is well written and the presented results and discussion are important for understanding the soil organic matter fate in changing Arctic regions. The authors should consider some minor comments before accepting the manuscript for publication with Biogeosciences.

Specific minor comments and suggestions:
- line 45: please check the reference (should be Frank et al., 2012)
- lines 58-59: the sentence should come before, the warming climate is already mentioned lines 38-39 for example
- lines 75-78: please clarify your objectives, the "physical fractionation" is an approach and not an objective
- line 81: add "Siberia" somewhere
- line 98: add "electric conductivity (EC)" to be consistent with line 117
- line 162: "to detect correlation": check the sentence, statistically a correlation is quantified and not "detected" using a plot
- line 179: the data could be presented with cumulative area charts for each profile and each element (C and N) to illustrate the proportions of each fraction by depth (in supplement)
- lines 221-222: move to the discussion
- line 271: "considerable amount of N" compared to what?
- line 283: change "C:N" to "C/N"
- line 340: change "dinitrogen" to "N2"
- figure 3: use "I" and "II" instead "a" and "b" to be consistent with the figure 2
- figure 4: the quality is too bad, be consistent with fig. 2-3, use indication of x log-scale and not just 10 and 100 that are not indicative, add minor gridlines for example), use the same color code as in figure 5
- figure 5: change the labels in the PCA to be in agreement with the text (C/N, a/o a ratio, etc.)
- figure 7: same comments as for the Excel plot in figure 4
- figure 8: I do not understand the point of adding both PCA and correlation matrices. I suggest to keep either the PCA, including individuals (as done in figure 5), or correlation matrices only

**Author's Response**

**Response to** *Interactive comment on* **"From fibrous plant residues to mineral-associated organic carbon – the fate of organic matter in Arctic permafrost soils"** *by* **Isabel Prater et al.** *from Anonymous Referee #1* **by the authors**

Dear Referee #1,

We thank you for your kind and helpful comments on our manuscript and we really appreciate that they helped to further improve it. Please find our answers below in *italics*, we also added the respective line numbers of the updated manuscript to improve the traceability:

- line 45: please check the reference (should be Frank et al., 2012)

*Thank you for this hint, the reference reads now Frank et al, 2012 (l. 51).*

- lines 58-59: the sentence should come before, the warming climate is already mentioned lines 38-39 for example

*As recommended, we moved the sentence (l. 43/44).*

- lines 75-78: please clarify your objectives, the "physical fractionation" is an approach and not an objective

*We rephrased the sentence to clarify that the detailed insights into chemical composition and stabilization mechanisms of SOM are the objectives and not the fractionation itself (l.80-86).*

- line 81: add "Siberia" somewhere

*We added "Siberian" (l. 89).*

- line 98: add "electric conductivity (EC)" to be consistent with line 117

*Thank you for the remark, we added (EC) (l. 109).*

- line 162: "to detect correlation": check the sentence, statistically a correlation is quantified and not "detected" using a plot

*We changed "detect correlations" to "identify interrelations" (l. 173).*

- line 179: the data could be presented with cumulative area charts for each profile and each element (C and N) to illustrate the proportions of each fraction by depth (in supplement)

*We thank you for this remark; it is always good to explore better ways to visualize data. We tested the cumulative approach, but decided to stick to the table as we think that a graphical depiction would be redundant.*

- lines 221-222: move to the discussion

*We moved this part of the sentence to l. 338/339.*

- line 271: "considerable amount of N" compared to what?

*We changed "considerable" to "noteworthy" (l. 284) as this expression works without comparison. We do not aim at comparing the values; we want to show that the N content should not be neglected.*

- line 283: change "C:N" to "C/N"

*Thank you for this hint, we changed it accordingly (l. 296).*

- line 340: change "dinitrogen" to "N2"

*To meet both this recommendation and the recommendations of the co-authors, we changed this part to "nitrous oxide ($N_2O$) and dinitrogen ($N_2$)" (l. 365).*

- figure 3: use "I" and "II" instead "a" and "b" to be consistent with the figure 2

*We reworked all our figures according to your suggestions and we appreciate that you helped to clearly improve them. We changed the captions and labels accordingly, used a consistent color code for all figures (that should also work for people with color vision deficiency) and took care of a much better quality.*

- figure 4: the quality is too bad, be consistent with fig. 2-3, use indication of x log-scale (and not just 10 and 100 that are not indicative, add minor gridlines for example), use the same color code as in figure 5

*We changed it accordingly, see comment above (figure 3).*

- figure 5: change the labels in the PCA to be in agreement with the text (C/N, a/o a ratio, etc.)

*We changed the figure, see comment above (figure 3).*

- figure 7: same comments as for the Excel plot in figure *4*

*We changed the figure accordingly, see comment above (figure 3).*

- figure 8: I do not understand the point of adding both PCA and correlation matrices. I suggest to keep either the PCA, including individuals (as done in figure 5), or correlation matrices only.

*We decided to follow your suggestion and left only the correlation matrices in the manuscript as we are convinced that those help best to illustrate the differences between large POM fractions and oPOMs fraction.*
The manuscript entitled "From fibrous plant residues to mineral-associated organic carbon – the fate of organic matter in Arctic permafrost soils" analysed four Cryosol soil cores for quantity and quality of organic matter to understand stabilization mechanisms and mineralization potential under climate change. The manuscript is very well written, within the scope of Biogeosciences and informative. No major methodological flaws were detected. The novelty of the results is somewhat limited, the results are close to what should have been expected. However, every new dataset on soils from these remote land masses that are warming up rapidly is valuable per se and the mix of methods is strong, of course. My most important major concern is that I was not able to understand how the authors exactly did the 1 m stock extrapolation when only some selected layers were fractionated. There is no information on that specific issue and looking at the reference publication reveals that soil profiles (usual for river terraces) were extremely heterogeneous with depth. This depth dependency and spatial heterogeneity in general, is of major importance for upscaling, which is again of specific importance for that vast and SOC rich region. This starts from how single soil profiles are averaged which should thus be done and described with care. A second major issue is related to the data shown in Fig. 4a: How do the authors explain this positive correlation of d13C and C/N ratio within the fractions, I think the opposite would be expected (the more decomposed, the more positive d13C)? Still, my recommendation is publish after some revisions. Please find specific comments below:

General: You might want to consider to call oPOM oPOMl (l for large), this would be more consistent when you also have oPOMs as individual fraction.
l.16: "the permafrost region" appears too unspecific. Is it the Northern Circumpolar Permafrost region or do you include the high mountain ranges here?
l.74: Why Ping et al. at the end?
l.94-97: Please be more specific about the sapling locations, or the selection of these 4 cores. Why those, any criteria to ensure that they are representative for the most likely very heterogeneous area? Can't really find information on that in the cited literature. Also, please mention the soil types. In Zubrzycki et al.2013, different soil types are mentioned. Was it all cryoturbated soils?
l.102: Which selected layers, how many samples were fractionated?
l.102: Table 1 is missing, but would be extremely important, also to judge about some interpretations of the authors.
l.117: During the washing... → How did you separate MAOM and oPOMs? Another density step? This is not clear.
l.126: Projected to 1m: How was this done exactly, and why is there no information about the depth distribution of fractions if the authors state that different depths were analysed? This is a bit confusing.
l.164: find correlations with what?
l.188: give ranges for the silt and sand-sized MAOM as well, otherwise the sentence reads incomplete.
l.217: This is something that is not clear to me: How do you explain this trend? Usually it should be the other way around. Why is d13C more negative when C/N ratios are decreasing?
Results in general: I was missing a depth distribution of the fractions, all results are depth independent and it is unclear, how homogeneous the profiles were.
l.268: It is not clear to me, if really cryoturbation caused the depth distribution of POM, but again this can only be judged if some depth information is included. Especially in river terraces, it could also be successive growth of the soil profile via sedimentation, and potentially even growth of organic layers.
l.286: Title sounds like POM fractions do also dominate N stocks, which doesn't seem to be the case (table 1). Maybe consider to rephrase.
l.293: "release vasts amount of N" → This is a contradiction to what has been said before and also the title (not much N in POM fractions)
l.313: Where does this information come from that fibres act as hot spots for microbial decay?
l.356: Where is this comparison per single soil layers?
Fig.4: X-axis: Why did you put the x-axis on top of the graph and show only 10 and 100? Is there any reason for that. Readability would improve when numbers are at the bottom and more continuous.

**Author's Response**

**Response to** *Interactive comment on* "From fibrous plant residues to mineral-associated organic carbon – the fate of organic matter in Arctic permafrost soils" *by* Isabel Prater et al. *from Anonymous Referee #2 by the authors*

Dear Referee #2,

We thank you for your valuable and very helpful comments on our manuscript! We appreciate that they supported us substantially with further improving it. Please find our answers to your remarks below in *italics*, we also added the respective line numbers of the updated manuscript to improve the traceability:

General: You might want to consider to call oPOM oPOMl (l for large), this would be more consistent when you also have oPOMs as individual fraction.

*We agree that your suggestion could imply some consistency in the naming. We discussed this and decided to keep our naming as the use of oPOMl and oPOMs could imply the existence of at least one more oPOM fraction between oPOMl and oPOMs and could therefore cause confusion. Furthermore, there are numerous publications that use the same terminology and we would like to maintain comparability for the reader.*

l.16: "the permafrost region" appears too unspecific. Is it the Northern Circumpolar Permafrost region or do you include the high mountain ranges here?

*We agree that this term is relatively broad. In l. 18 we mentioned the Arctic and according to your suggestion we additionally added "Northern circumpolar" to avoid further misunderstandings (l. 19).*

l.74: Why Ping et al. at the end?

*We placed Ping et al. 2015 here (l. 79), as we think this paper provides a great overview on the facts that we stated before.*

l.94-97: Please be more specific about the sapling locations, or the selection of these 4 cores. Why those, any criteria to ensure that they are representative for the most likely very heterogeneous area? Can't really find information on that in the cited literature. Also, please mention the soil types. In Zubrzycki et al.2013, different soil types are mentioned. Was it all cryoturbated soils?

*You are right that this is a very important point in our study. We further specified the sampling area (l. 104), and referred here directly to the Holocene river terrace and also added another reference with more details on the sampling area (l. 108). We also added information on the soils (l. 93-95).*

l.102: Which selected layers, how many samples were fractionated?
l.102: Table 1 is missing, but would be extremely important, also to judge about some interpretations of the authors.

*We thank you very much for this crucial hint. We moved the respective table to the Supplement, but at this point (l.113) we did not change the reference. The information on the samples as the depth layers etc. is now given in table S1. We also added the number of the selected layers we fractionated (l. 113).*

l.117: During the washing... → How did you separate MAOM and oPOMs? Another density step? This is not clear.

*As this step is very fundamental for our study we are thankful for your remark. To better clarify the procedure, we added more detailed information (l. 128/129). We followed a standard density fractionation approach, where we separated the POM fractions (including oPOMs) from the MAOM by density fractionation, which we describe in l. 116-129. The oPOMs was not separated from the MAOM, but from the oPOM fraction as described in l. 128/129.*

l.126: Projected to 1m: How was this done exactly, and why is there no information about the depth distribution of fractions if the authors state that different depths were analysed? This is a bit confusing.

*The stocks for the respective sampled and analyzed soil depths were taken as a bulk and projected to one cubic meter. As common practice, we report the stocks based on the sampled and analyzed material. The depth distribution and all according information can be found in table S1.*

l.164: find correlations with what?

*We rephrased this sentence (l. 173) to better clarify the statement.*

l.188: give ranges for the silt and sand-sized MAOM as well, otherwise the sentence reads incomplete.

*Thank you, we totally agree and added the missing information (l. 200/201).*

l.217: This is something that is not clear to me: How do you explain this trend? Usually it should be the other way around. Why is d13C more negative when C/N ratios are decreasing?

*You are pointing to a very interesting aspect of our work that nicely demonstrates the specificity of the studied soil systems. We discuss this in more detail in the discussion section 4.3 (l. 338-356). We rewrote this section to better emphasize the differences in the $^{13}$C abundance for different SOM fractions.*

Results in general: I was missing a depth distribution of the fractions, all results are depth independent and it is unclear, how homogeneous the profiles were.

*You are right that the depth distribution of fractions is an important fact. We are now providing all according information in table S1.*

l.268: It is not clear to me, if really cryoturbation caused the depth distribution of POM, but again this can only be judged if some depth information is included. Especially in river terraces, it could also be successive growth of the soil profile via sedimentation, and potentially even growth of organic layers.

*You are right, in areas that are flooded regularly a burial of organic matter can be expected. This was the reason we avoided such areas and did not take samples from the floodplain. We only sampled the terrace that is only very rarely flooded, which is thought not to lead to the burial of pockets of organic matter as the ones discovered. Related information on the depth distribution is given in table S1.*

l.286: Title sounds like POM fractions do also dominate N stocks, which doesn't seem to be the case (table 1). Maybe consider to rephrase.

*In the studied soils, the particulate OM represents an important N storage pool. While the C stock is dominated by the large POM fractions, the N stock is dominated by the fPOM and the oPOMs fraction – especially the fPOM fraction plays a crucial role for both stocks. We discuss this in l. 302-306.*

l.293: "release vasts amount of N" → This is a contradiction to what has been said before and also the title (not much N in POM fractions)

*We found relatively high amounts of N in fPOM, oPOMs and clay-sized MAOM, therefore, we assume an increased release of this N under ongoing warming. With the title we want to indicate that the POM fractions are more important for the C stock than for the N stock, not that the POM fractions are negligible.*

l.313: Where does this information come from that fibres act as hot spots for microbial decay?

*Thank you for this hint, we added a source for this information (l. 329) and rewrote parts of the respective section. The fibrous OM particles represent relatively undecomposed plant residues, as we were able to demonstrate using NMR spectroscopy. Thus, these particles represent detrital material*

*that most likely provides highly bioavailable OM sources for microbial activity. This hot spot effect of the detritusphere for microbial activity is well known and was addressed in numerous other studies (e.g. Beare et al. 1995, Poll et al. 2006, 2008, Sanaullah et al. 2016).*

l.356: Where is this comparison per single soil layers?

*Thank you for this hint, we added the respective information (l. 382).*

Fig.4: X-axis: Why did you put the x-axis on top of the graph and show only 10 and 100? Is there any reason for that. Readability would improve when numbers are at the bottom and more continuous.

*Thank you for this remark, we reworked all of our figures and changed fig. 4 according to your suggestion.*
This is a very well-written manuscript that describes organic matter content and composition of physically-isolated density and particle size fractions collected from ice-wedge polygon centers in the Arctic. The objective of the paper is to characterize degree of decomposition of organic matter in permafrost soils with varying degrees of association with mineral surfaces to better understand potential bioavailability of this organic matter pool to warming and thawing. The authors present a thorough chemical characterization of particulate and mineral associated organic matter pools through C and N elemental analysis, stable isotopes and C13-NMR spectroscopy. The results interestingly reveal large contributions of potentially chemically bioavailable POM to the bulk soil C pool, whereas mineral-associated fractions contribute more to the soil N pool. This work has implications for predictions of the response of similar permafrost-affected soils to warming.

Abstract:
L. 25: "We demonstrate that" It would be helpful in this sentence to operationally identify the fraction being discussed (that is, how was it isolated physically?) to better understand how it is being interpreted as "bioaccessible." Can you define the term bioaccessible? Is it synonymous with the more common "bioavailable" or does it specifically refer to physical accessibility?
Methods: The methods indicate soil drill cores are taken but do not highlight what depths are analyzed and presented. The text states in L. 102 : "Our analyses focused on selected layers only, as shown in Table 1" but Table 1 does not include this information. One would expect that the contribution of POM vs MAOM and the state of decomposition may vary with soil depth (perhaps not in the traditional predictions) yet the paper does not describe what depths are being analyzed.
Discussion:
The discussion is quite long with extensive paragraphs that have multiple ideas, which makes it sometimes a little difficult to follow all the ideas. Consider where the discussion can be streamlined and how paragraphs could be split into smaller blocks of text.
Section 4.1- The section heading is perhaps not the most informative of the text, as permafrost processes (other than one mention to cryoturbation) are not discussed in depth here. Consider renaming the section or including more information on processes. It may also be helpful to separate the text into a paragraph on C and N stocks and another one on composition of SOM, mainly C:N ratios.
Section 4.2: It is very interesting that the POM and MAOM fractions play such different roles in C and N storage in these soils.
Section 4.3: Consider starting the paragraph l. 332 with summarizing results of N dynamics or 15N and their implication as the first sentence on N fixation seems to have no context. This paragraph could also be moved after the NMR paragraph which flows better after the 13C paragraph.
Minor edits:
Introduction, paragraph starting l. 58-78 is too long with too may different ideas. Should be broken up into smaller paragraphs, one on effects of climate change on SOM, one on SOM methods, then the research objectives.
Spell out abbreviations for symbols in the Table legends. For example, fPOM, MAOM...
Also indicate whether data reported are means and standard error or means and standard deviation.
Table 2. Should a/o-a ratio be O-a ratio? (capital O)
Figure 1. May be helpful to indicate what the white and blue colors are on the image.
Ice and open water? Unclear because the ocean is black.
l. 240: add ppm after 70-75 ppm /52-57 ppm

Author's Response

**Response to *Interactive comment on* "From fibrous plant residues to mineral-associated organic carbon – the fate of organic matter in Arctic permafrost soils" *by* Isabel Prater et al. *from Anonymous Referee #3* by the authors**

Dear Referee #3,

We are very grateful for your helpful and supporting comments on our manuscript that help to further improve it. Please find our answers to your remarks below in *italics*, we also added the respective line numbers of the updated manuscript to improve the traceability:

L. 25: "We demonstrate that" It would be helpful in this sentence to operationally identify the fraction being discussed (that is, how was it isolated physically?) to better understand how it is being interpreted as "bioaccessible." Can you define the term bioaccessible? Is it synonymous with the more common "bioavailable" or does it specifically refer to physical accessibility?

*We agree that these terms are often used in a confusing way. When we aim at emphasizing the spatial inaccessibility of the OM, we use "bioaccessibility". When we refer to the microbial availability determined by the chemical composition of a substrate, we use "bioavailability". We made some changes in our manuscript according to our remark.*

Methods: The methods indicate soil drill cores are taken but do not highlight what depths are analyzed and presented. The text states in L. 102 : "Our analyses focused on selected layers only, as shown in Table 1" but Table 1 does not include this information. One would expect that the contribution of POM vs MAOM and the state of decomposition may vary with soil depth (perhaps not in the traditional predictions) yet the paper does not describe what depths are being analyzed.

*We thank you very much for this crucial hint that we were not aware of. We moved the respective table to the Supplement, but at this point (l. 113) we did not change the reference. The information on the samples like depth layers etc. is now given in table S1 and we corrected the reference accordingly.*

Discussion:
The discussion is quite long with extensive paragraphs that have multiple ideas, which makes it sometimes a little difficult to follow all the ideas. Consider where the discussion can be streamlined and how paragraphs could be split into smaller blocks of text.

*According to your suggestion, we restructured the discussion to increase the readability of the manuscript and split the paragraph discussing stable isotopes and NMR results into two paragraphs.*

Section 4.1: The section heading is perhaps not the most informative of the text, as permafrost processes (other than one mention to cryoturbation) are not discussed in depth here. Consider renaming the section or including more information on processes. It may also be helpful to separate the text into a paragraph on C and N stocks and another one on composition of SOM, mainly C:N ratios.

*According to your suggestion we changed the heading to "Cryoturbation determines bulk soil organic matter distribution". As we do not widely discuss the C and N stocks, the aim of the manuscript is clearly on the composition of the SOM fractions, thus we would like to stick to the current paragraph.*

Section 4.2: It is very interesting that the POM and MAOM fractions play such different roles in C and N storage in these soils.

*We are happy that you acknowledge that this is an interesting finding in our study.*

Section 4.3: Consider starting the paragraph l. 332 with summarizing results of N dynamics or 15N and their implication as the first sentence on N fixation seems to have no context. This paragraph could also be moved after the NMR paragraph which flows better after the 13C paragraph.

*We followed your suggestion above and separated the paragraph further. We have now one paragraph (4.3) discussing d13C and d15N and we slightly rearranged this paragraph. Another paragraph (4.4) is now only focusing on the NMR discussion.*

Minor edits:
Introduction, paragraph starting l. 58-78 is too long with too may different ideas. Should be broken up into smaller paragraphs, one on effects of climate change on SOM, one on SOM methods, then the research objectives.

*We slightly restructured the Introduction according to your suggestion.*

Spell out abbreviations for symbols in the Table legends. For example, fPOM, MAOM... Also indicate whether data reported are means and standard error or means and standard deviation.

*Thank you for this remark, we added the missing information to the captions of the tables.*

Table 2. Should a/o-a ratio be O-a ratio? (capital O)

*This ratio relates to functional groups that consist of O/N-alkyl-C and Alkyl-C, which is normally given as "Alkyl-C to O/N-alkyl-C ratio". To make it easier to read, we defined the ratio of alkyl C to O/N alkyl C as a/o-a ratio in the method section (2.4) and we kept this wording throughout the manuscript and in the tables and figures as well.*

Figure 1. May be helpful to indicate what the white and blue colors are on the image.
Ice and open water? Unclear because the ocean is black.

*Thank you for this important remark, we added the information according to your suggestion. The white color mainly in the western part is the unvegetated sandy sediment of the floodplain and the blue spots indicate water: larger water bodies and shallow water on the terrace.*

l. 240: add ppm after 70-75 ppm /52-57 ppm

*We introduce the ratio according to Bonanomi et al. (2013) in the methods section (2.4), where we clearly state the chemical shift regions that are considered for this decomposition proxy. To increase readability we use a reduced naming of the ratio which is in accordance with the "a/o-a ratio" term. Thus, we would like to be consistent in the form that we use to express NMR-derived decomposition proxies.*
The manuscript by Prater et al. provides new data on the different physical fractions of soil organic matter from the Lena River Delta in the Arctic. The area on Samoylov Island is characterized by permafrost. The authors investigated soils with respect to the composition and distribution of organic C among differently stabilized SOM fractions, in order to gain knowledge on the mechanisms stabilizing organic C in Arctic soils, besides impaired decomposition due to low temperatures. The methods consists of the use of sophisticated approaches, separating SOM into different fractions, allowing for a detailed understanding of the stabilization mechanisms of organic carbon in soils. The study is relevant, as there are still rather few analytical approaches to the stabilization mechanisms to assess the variability of C stocks in tundra soils. The research question is particularly relevant as the study deals with a region where permafrost occurs and where soils are both an important store of carbon and other greenhouse gases and are affected by global warming. The authors did not formulate a clear hypothesis or the expected results. Consequently, they did not make a comparison between their results and their original expectations, which makes it difficult to compare the results of the research with other studies.

General comments and suggestions:
- The research question addressed by the authors is important due to the lack of knowledge on the topic. Indeed, researchers only recently started to understand the importance of cold soils for the global carbon cycle, and thus global climate. As a consequence, only a few studies related to this topic have been made so far. - The authors did not explicitly state any hypotheses. They described their intent of investigating the effect of climate change on the carbon stabilization in permafrost-affected soils, but they remained vague and did not state any kind of expected results. Therefore, it is difficult to understand to what extent the research contributed to their question. – The study site is situated in the river delta of the River Lena. Chemical composition and structure of the soil could be the result of flooding which is not the case for typical arctic permafrost soils. In general, the isle may be more affected by the Lena itself than by the rising temperature. In addition, the closeness of the Siberian sea will have an influence of the isle too, as the ocean moderates the temperatures. Therefore, the study site on the isle Samoylov maybe not representative for arctic permafrost soils in general. - Do you think is the d15N a suitable method? There are many uncertainties related to it, which could be elaborated upon.

Specific comments and suggestions:
L. 75-78: Here the authors write about their approach and the aims, which are basically to gain better knowledge on the topic. Since this section is at the end of the introduction, we think that this part is the most suited for adding the research questions and hypotheses. We think this is important, especially because the authors took four soil cores in a vast area that might be highly heterogeneous. Therefore, having expectations related to the SOM fractions you expect to find in this area, including also the stratification of the soil layers could help determining how representative the four soil cores are with respect to the whole study area.
L. 94-97: In this part the methods are described. However, the authors then state that "a detailed description of the study area and the sampling of the soil cores can be found in Zubrzycki et al. (2013)".We advise that the authors include all relevant information also in the presented manuscript. Otherwise, the readers have to go into the literature to find this relevant information.
L. 101-105: Here, the authors describe how samples were collected, but omitted to state how many samples were collected for each SOM fraction and from which soil core they were collected. We advise to provide the number of samples of each SOM fractionation type, because otherwise it might be difficult to interpret the graphs. We also checked the literature but found any information about the number of samples in Zubrzycki et al., 2013.
Fig. 4: The three graphics (figure 4. a, b, c) could be made more similar. Further, for what concerns figure a and b, the authors represented only the two extreme values on the x-axis (10 and 100), which makes it difficult to infer the values of the dots in the middle of the graph. Please include more labels on the x-axis to make it more continuous and improve readability. We would also advice to put the x-axis on the bottom for both graphs (a and b) and not once on the top and once on the bottom.
Further, we noticed a clear positive correlation between the C/N ratio and the d13C, however, since the C/N ratio usually decreases during ongoing decomposition, we were expecting the opposite trend. We therefore advise to further explain the meaning of this positive correlation.
Fig.7 & 8: Graphs 7 and 8 are difficult to interpret and would require more information in the captions to make the graphs understandable without the reader having to look up more information in the main text.
Title: Why was the word "fibrous" included in the title? Almost all plants residues are fibrous, except for plant exudates. Do you specifically looked at fibrous plant residues omitting exudates? Further, the fate of organic matter sounds somewhat dramatic.
We think the title could be shortened to, for example: "From plant residues to mineralassociated organic carbon in Arctic permafrost soils".

**Author's Response**

**Response to *Interactive comment on* "From fibrous plant residues to mineral-associated organic carbon – the fate of organic matter in Arctic permafrost soils" *by* Isabel Prater et al. *from Marijn Van de Broek and students* by the authors**

Dear master students,

Thank you for your helpful comments and for the effort you put into your review. And congratulations that you had the chance to prepare reviews as exercise during a seminar – that is a really useful training. We appreciate your comments; please find our answers below in *italics*:

- The research question addressed by the authors is important due to the lack of knowledge on the topic. Indeed, researchers only recently started to understand the importance of cold soils for the global carbon cycle, and thus global climate. As a consequence, only a few studies related to this topic have been made so far.

*We are happy that you share our view on the current rather sparse knowledge on the topic of our research.*

- The authors did not explicitly state any hypotheses. They described their intent of investigating the effect of climate change on the carbon stabilization in permafrost-affected soils, but they remained vague and did not state any kind of expected results. Therefore, it is difficult to understand to what extent the research contributed to their question.

*We are happy that you raised this issue, we now better clarified our objectives and added our expectations in the final part of the introduction.*

– The study site is situated in the river delta of the River Lena. Chemical composition and structure of the soil could be the result of flooding which is not the case for typical arctic permafrost soils. In general, the isle may be more affected by the Lena itself than by the rising temperature. In addition, the closeness of the Siberian sea will have an influence of the isle too, as the ocean moderates the temperatures. Therefore, the study site on the isle Samoylov maybe not representative for arctic permafrost soils in general.

*As described in section "2.1 Site characteristics and soil sampling", we took the samples from the Holocene river terrace that is rarely flooded. To avoid the impact of regular flooding, we did not take samples from the active floodplain. Your remark regarding the influence of the river and the Laptev Sea is very important. We address the differences between the climate on the island and on the mainland in our site description. However, the studied Cryosol types can be found throughout the Arctic and thus we assume the demonstrated properties and suggested processes can be applied in a general sense.*

- Do you think is the d15N a suitable method? There are many uncertainties related to it, which could be elaborated upon.

*The use of d15N as an integrator of the N cycle is a semi-quantitative method, which does not allow to derive quantitative process information. Thus, we agree that this method requires a careful discussion of the various processes/fractionation factors and isotopic signatures of N sources that jointly determine d15N, which is precisely what we do in this manuscript. Taking this into account, it is a very powerful tool to fingerprint dominating N cycle processes and general N cycle patterns (e.g., open N cycle, closed N cycle) without the need for disturbance by e.g., adding enriched isotope tracers. In this context, it is also important to relate d15N to other measured parameters to support its interpretation, which is what we do in this study as well.*

Specific comments and suggestions:
L. 75-78: Here the authors write about their approach and the aims, which are basically to gain better knowledge on the topic. Since this section is at the end of the introduction, we think that this part is the most suited for adding the research questions and hypotheses. We think this is important, especially because the authors took four soil cores in a vast area that might be highly heterogeneous. Therefore, having expectations related to the SOM fractions you expect to find in this area, including also the stratification of the soil layers could help determining how representative the four soil cores are with respect to the whole study area.

*We agree that hypotheses are one possible way to communicate expectations at the start of a study. We have added a sentence stating what we generally expected to find without naming this explicitly as hypothesis. The cores were taken from a Holocene river terrace and were cryoturbated. We are giving the detailed information on the soil cores in table S1 in the supplement. As we sampled in depth layers, we have no particular information about the stratification, however, there were no indications of flooding events that might have resulted in stratigraphic shifts.*

L. 94-97: In this part the methods are described. However, the authors then state that "a detailed description of the study area and the sampling of the soil cores can be found in Zubrzycki et al. (2013)".We advise that the authors include all relevant information also in the presented manuscript. Otherwise, the readers have to go into the literature to find this relevant information.

*Thank you for this remark; we added more information in the text. As we did not want to overload this section, we focused on the information that is relevant for our study. For readers who want to have more information, we added the respective references.*

L. 101-105: Here, the authors describe how samples were collected, but omitted to state how many samples were collected for each SOM fraction and from which soil core they were collected. We advise to provide the number of samples of each SOM fractionation type, because otherwise it might be difficult to interpret the graphs. We also checked the literature but found any information about the number of samples in Zubrzycki et al., 2013.

*Thank you for this very important remark. A detailed list of the samples and their properties can be found in table S1 and we added this information (l. 113). In addition, we have to excuse that we had a mistake in the reference (it said table 1 instead of table S1 before). We also added the number of selected layers in the same line.*

Fig. 4: The three graphics (figure 4. a, b, c) could be made more similar. Further, for what concerns figure a and b, the authors represented only the two extreme values C3on the x-axis (10 and 100), which makes it difficult to infer the values of the dots in the middle of the graph. Please include more labels on the x-axis to make it more continuous and improve readability. We would also advice to put the x-axis on the bottom for both graphs (a and b) and not once on the top and once on the bottom. Further, we noticed a clear positive correlation between the C/N ratio and the d13C, however, since the C/N ratio usually decreases during ongoing decomposition, we were expecting the opposite trend. We therefore advise to further explain the meaning of this positive correlation.

*Thank you for this hint. We reworked all of our figures and made them easier to read and took care of a better quality. We also implemented a color scheme for all figures, changed the values on the x-axis*

*and put all x-axes to the bottom. Regarding d13C and C/N ratio, we rewrote the respective section (4.3) to better emphasize the differences between the SOM fractions.*

Fig.7 & 8: Graphs 7 and 8 are difficult to interpret and would require more information in the captions to make the graphs understandable without the reader having to look up more information in the main text.

*We reworked both figures and think they are easier to read and understand now.*

Title: Why was the word "fibrous" included in the title? Almost all plants residues are fibrous, except for plant exudates. Do you specifically looked at fibrous plant residues omitting exudates? Further, the fate of organic matter sounds somewhat dramatic. We think the title could be shortened to, for example: "From plant residues to mineralassociated organic carbon in Arctic permafrost soils".

*We use "fibrous" because we see a clear difference between the large, fibrous POM fractions and the oPOMs fraction that is not fibrous anymore – we describe this at length, but the difference becomes especially obvious in figure 9. We want to emphasize the specific macroscopic nature of the cryoturbated materials referring to the occurrence that one is experiencing when sampling. And the fibrous litter residues foster the bulky soil structure that drives the specific oPOMs formation, which is clearly different from less organic temperate soils. We want to use the word "fate" as it summarizes a development from initial litter residues to more transformed soil compartments, which we think best describes the partitioning of compounds to various processes or pools/compounds in biogeochemistry.*

**Author's changes in manuscript**

Please find below our reworked manuscript with changes highlighted in yellow:

[revised manuscript text omitted]